# Fluoropolymer Membranes for Membrane Distillation and Membrane Crystallization

**DOI:** 10.3390/polym14245439

**Published:** 2022-12-12

**Authors:** Xue Li, Jun Pan, Francesca Macedonio, Claudia Ursino, Mauro Carraro, Marcella Bonchio, Enrico Drioli, Alberto Figoli, Zhaohui Wang, Zhaoliang Cui

**Affiliations:** 1Institute on Membrane Technology, ITM-CNR, Via P. Bucci 17/C, 87036 Arcavacata di Rende, Italy; 2Department of Chemical Science, University of Padova, Via F. Marzolo 1, 35131 Padova, Italy; 3State Key Laboratory of Materials-Oriented Chemical Engineering, College of Chemical Engineering, Nanjing Tech University, Nanjing 210009, China; 4National Engineering Research Center for Special Separation Membrane, Nanjing Tech University, Nanjing 210009, China

**Keywords:** membrane distillation, membrane crystallization, fluoropolymer membranes, hydrophobic modification

## Abstract

Fluoropolymer membranes are applied in membrane operations such as membrane distillation and membrane crystallization where hydrophobic porous membranes act as a physical barrier separating two phases. Due to their hydrophobic nature, only gaseous molecules are allowed to pass through the membrane and are collected on the permeate side, while the aqueous solution cannot penetrate. However, these two processes suffer problems such as membrane wetting, fouling or scaling. Membrane wetting is a common and undesired phenomenon, which is caused by the loss of hydrophobicity of the porous membrane employed. This greatly affects the mass transfer efficiency and separation efficiency. Simultaneously, membrane fouling occurs, along with membrane wetting and scaling, which greatly reduces the lifespan of the membranes. Therefore, strategies to improve the hydrophobicity of membranes have been widely investigated by researchers. In this direction, hydrophobic fluoropolymer membrane materials are employed more and more for membrane distillation and membrane crystallization thanks to their high chemical and thermal resistance. This paper summarizes different preparation methods of these fluoropolymer membrane, such as non-solvent-induced phase separation (NIPS), thermally-induced phase separation (TIPS), vapor-induced phase separation (VIPS), etc. Hydrophobic modification methods, including surface coating, surface grafting and blending, etc., are also introduced. Moreover, the research advances on the application of less toxic solvents for preparing these membranes are herein reviewed. This review aims to provide guidance to researchers for their future membrane development in membrane distillation and membrane crystallization, using fluoropolymer materials.

## 1. Introduction

In recent years, membrane technologies have been predominantly applied in many fields, such as food, pharmaceutical, printing, papermaking, aerospace, etc. This is due to their strong separation ability and many other advantages such as low cost, energy-saving and environmental impact compared to conventional technologies. Popular membrane operations include microfiltration (MF), ultrafiltration (UF), nanofiltration (NF), reverse osmosis (RO), membrane distillation (MD), electrodialysis (ED), membrane contactors, etc. Among all these membrane technologies, MD and MCr are two membrane processes belonging to the membrane contactors family which need the use of microporous hydrophobic membranes. In these two processes, the microporous hydrophobic membrane acts as the interface to separate two phases, and the membrane pores provide the transport channels to realize mass transfer between the two phases. Different from the selective separation function of other general membrane processes, in the membrane contactors (as MD and MCr) the membrane does not have any selectivity for each component, but only acts as a barrier between two phases: the liquid phase is in contact with the feed side of the membrane, and the vapor phase is in contact with the permeate side of the membrane. Only volatile components can transfer from one phase to another through the membrane pores and are finally collected in the permeate side. The hydrophobic nature of the membrane prevents the feed liquid from penetrating into the membrane pores. Therefore, all non-volatile components are rejected in the feed side. In principle, the rejection of MD and MCr can reach 100%. The driving force of MD and MCr is the vapor pressure difference caused by the temperature difference and/or pressure difference (i.e., temperature difference for direct contact membrane distillation (DCMD) or pressure difference for vacuum membrane distillation (VMD)).

Main features of MD and MCr processes are shown in Table 1.

During the operation of these two membrane processes, the most common problems are membrane fouling and wetting. Due to the hydrophobicity of the membrane, the feed liquid cannot pass through the membrane pores, and only the gas molecules can pass through, being finally collected on the permeate side of the membrane. However, once the hydrophobic membrane begins to get wet, liquid begins to enter the membrane pores. Firstly, the surface is wetted and partial pores are wetted. Then, most of membrane pores get wet which results in a decrease of separation efficiency of the membrane, as shown in Figure 1 [13].

Wetting phenomenon can be classified into three types: surface wetting, partial wetting and fully wetting. When only the membrane surface is wetted, the membrane does not lose its separation function; when partial membrane pores are wetted, the membrane starts to lose its separation ability and the conductivity of the permeate increases gradually. When all the membrane pores are wetted, the membrane loses its separation ability completely. At the same time, considering the existence of different pollutants (micro particles, acid, base, oil, etc.) in the feed solution, this will lead to the deposition of pollutants on the membrane surface or in the membrane pores, and also destroying the membrane surface structure and reducing the hydrophobicity of the membrane. This is typical membrane fouling. The latter is another undesired phenomenon that can severely degrade membrane performance and negatively impact the membranes during long-term operation. Even if it has been studied and investigated by researchers for many years, its mechanism is not well understood yet [14,15,16]. In general, membrane fouling can be divided into organic fouling, inorganic fouling and biological fouling based on the classification of foulant types. Inorganic fouling, or scaling, includes alkaline scaling, non-alkaline scaling and uncharged molecule scaling. The most common alkaline scaling is CaCO_3_. It can be formed from different feeds such as seawater, groundwater or brackish water disposal. It can also be generated from the breakdown of bicarbonate. It is reported that a temperature of 37 °C can be considered as the minimum temperature for the formation of CaCO_3_ from the mildly concentrated ocean water. Other inorganic scales or colloidal fouling are Mg(OH)_2_, CaSO_4_, CaSO_4_•H_2_O, CaPO_4_, MgCl_2_, silica, Fe_2_O_3_, Fe(OH)_3_, Fe_3_O_4_, FeO, etc. Organic fouling is defined as the collection of carbon-based material on a membrane. Biofouling, normally occurs through food, beverage, and industrial wastewater as feed, of which their COD are very high. It includes bacteria, fungi, etc. Fouling images on the membrane surface in MD are shown in the following Figure 2.

Figure 2 shows different foulants on the membrane surface. Fouling can also be classified into reversible and irreversible fouling according to the degree of its cleaning recovery [20]. Figure 3 shows three different mechanisms of fouling: cake layer fouling, pore constriction and pore blockage. The first two types are reversible fouling since the cake layer and the adsorption of foulants on the pore surface can be cleaned by suitable cleaning methods such as backwashing, etc. Conversely, when pore blockage occurs, it cannot be cleaned completely. Therefore, pore blockage can be regarded as irreversible fouling [21]. The existence of membrane fouling can increase trans-membrane resistance and decrease membrane flux. The formation of the cake layer or the biofilm on the membrane surface can increase the temperature polarization phenomenon and decrease the separation performance. Fouling is influenced by the characteristics of feeds (pH, COD, etc.), operation conditions (temperature, flow rate, etc.), cleaning method and most importantly the properties of the membrane. Regardless of the unchangeable conditions of feeds, the studies to improve the membrane hydrophobicity and ensure its anti-fouling/wetting performance during the processes have gradually become one of the key issues of many researches. One way to improve membrane anti-fouling/wetting properties is to choose proper hydrophobic membrane materials and make hydrophobic modifications of the membrane surface.

Typical polymeric membrane materials used for MD and MCr processes are polypropylene (PP) and fluoropolymer materials such as polyvinylidene fluoride (PVDF) and its copolymers, poly(tetrafluoroethylene) (PTFE), etc. [22,23]. Compared with other polymeric materials, fluoropolymer materials are attracting more and more attention in the field of membrane preparation due to their excellent properties, such as superior thermal and chemical resistance. Fluorine (F) atoms have high electronegativity and low polarizability, which can form strong C-F bonds, endow polymers with good thermal and chemical stability and low surface energy. The characteristics of C-F bond offers unique properties because of its high bond energy of 485 kJ/mol [24]. C-X bond energy data are reported in literature for the methane model molecules as CH_3_-X, CH_2_F-X, CHF_2_-X and CF_3_-X, where X = H, F, Cl, or Br. The corresponding bond energies are shown in the following Figure 4, which shows that the C-F bond contains more energy than other C-H, C-Cl and C-Br bonds.

The higher the number of C-F bonds, the higher the bond energy of the compound. Fluorous monomers include tetrafluoroethylene (TFE), hexafluoropropylene (HFP), vinylidenefluoride (VDF), and chlorotrifluoroethylene (CTFE), etc. Different combinations of these monomers or copolymers could generate different compounds: PVDF, PTFE, poly(ethylene chlorotrifluoroethylene) (ECTFE), poly(ethylene-alt-tetrafluoroethylene) (ETFE), poly(tetrafluoroethylene-co-hexafluoropylene) (FEP), poly(chlorotrifluoroethylene) (PCTFE), polyvinyl fluoride (PVF), poly(tetrafluoroethylene-co-perfluoropropyl vinyl ether) (PFA), and poly(tetrafluoroethylene-co-hexafluoropylene-vinylidene fluoride) (THV), etc. [25,26]. The number of publications of fluoropolymer membranes applied in MD and MCr processes in the past 10 years is shown in Figure 5 below. It has gradually increased from around 300 to about 450 in the recent decade, showing an active interest in related fields.

The review structurally reports the details on membrane distillation and membrane crystallization processes and the fluoropolymer membranes usually utilized. The preparation methods of fluoropolymer membranes and their modification methods are also reviewed. Among all the membrane preparation techniques, non-solvent-induced phase separation (NIPS), thermally-induced phase separation (TIPS) and vapor-induced phase separation (VIPS) are described in detail. Membrane modification methods include surface coating method, grafting method and blending method. In addition, some new membrane preparation and modification methods are also being explored and summarized, including membrane preparation with greener solvents [27,28].

## 2. Membrane Distillation and Membrane Crystallization

### 2.1. Membrane Distillation (MD)

As mentioned above, MD is a thermally driven membrane separation process with a porous hydrophobic membrane as the separation medium, and the vapor pressure difference on both sides of the membrane as the driving force (the principle of the process is shown in Figure 6). One side of the hydrophobic membrane is in contact with the hot feed solution. As the volatile components in feed solution pass through the membrane pores, they are condensed and collected by the cooling medium on the permeate side. The main configurations of MD are direct contact membrane distillation (DCMD), vacuum membrane distillation (VMD), air gap membrane distillation (AGMD) and sweeping gas membrane distillation (SGMD), as shown in Figure 7. They differ according to the cooling medium on the permeate side.

DCMD is normally defined as the simplest design; here, the cooling liquid and the hot feed are directly in contact with both sides of the membrane, so the driving force is the vapor pressure difference caused by the temperature difference. Most DCMD configurations adopt counter-current configurations due to the maximum heat transfer efficiency and thus leading to the maximum mass transfer [29]. The flow rate of the hot side is normally higher than that in the cooling side. If the permeate side is changed into a vacuum, the configuration becomes VMD. The driving force is caused by the pressure difference between two sides of the membrane. Due to the constant vacuuming, the pressure for the permeate side is regarded as extremely low, so the driving force is higher compared with other types of configurations. For AGMD, there is an air gap in the permeate side and the vapor passes through the membrane as well as the air gap, and finally condenses on the cold surface of the permeate side. The transfer resistance for this type of configuration is high due to the existence of the air gap. Another configuration is SGMD, where an inert gas is normally selected as the cold sweep gas (such as nitrogen (N_2_), carbon dioxide (CO_2_), etc.). There are also other classifications of MD, such as osmosis membrane distillation (OMD), which is similar to DCMD but the permeate side contains a stripping solution, such as NaCl, MgCl_2_, or CaCl_2_ solution. The concentration of the stripping solution acts as the partial pressure contribution for the driving force. Liquid gap membrane distillation (LGMD) is one of the most recent configurations where a cold liquid (usually pure water) is normally maintained between a membrane and a cold surface. This configuration is rarely mentioned and applied in the literature, so it is not classified as a common configuration. Similarly, thermostatic sweeping gas membrane distillation (TSGMD) is a combination of SGMD and AGMD, of which a cold inert gas is circulated through the permeate side, carrying out the vapor permeate and condensing outside the membrane module in a condenser [30]. Other configurations such as vacuum–air gap membrane distillation (V-AGMD), vacuum-multi-effect membrane distillation (V-MEMD), permeate gap membrane distillation (PGMD) and material gap membrane distillation (MGMD) are also used as MD configurations.

The characteristics of these MD configurations are different in terms of energy efficiency, vapor flux, condensation method and thermal energy recovery. DCMD configuration is the simplest configuration and the one with the most choices in the experiments. DCMD configuration is characterized by high trans-membrane flux but lower thermal energy efficiency, while AGMD configuration presents high thermal energy efficiency but low flux [31]. AGMD configuration is more considered than SGMD configuration, but the problem of AGMD is that it is more complex to build up and the air gap is controlled with difficulty, which may induce the permeate accumulation in the air gap side [32]. Membranes for VMD configuration suffer the most severe fouling problem and have the highest wetting potential due to its higher pressure difference. In addition, the energy recovering of the SGMD and VMD configurations is more difficult than DCMD and AGMD configurations since the latter two configurations can recover the energy from the cooling plate and the permeate, respectively [33].

Electrical energy consumption in MD process is low because the pressure used is very low (around one bar) compared with pressure-driven processes such as RO. The recovery factor of MD is also high and it suffers the lowest concentration polarization phenomenon with respect to RO. The temperature used for the feed solution is relatively low and does not need to reach boiling temperature. So, it can be compatible with low-grade heat (such as waste heat) or sustainable heat (such as solar resources). It is independent with salinity; therefore, it can be used to treat highly concentrated solutions such as highly concentrated RO brine or high-salinity wastewater [34]. For what concerns the MD plant, it can be more compact and takes up less space compared to conventional distillation designs.

Although MD has many advantages, it also has disadvantages, such as temperature polarization (TP) and wetting. TP is the combined effect of fluid dynamics, low trans-membrane flux and heat necessary for the evaporation of the feed that reduces the temperature on the membrane surface [35].

The most significant problem of MD is that it still has limited applications at the industrial level due to the lack of available membranes. The membrane should have characteristics that meet the MD requirements. It should normally be hydrophobic and micro-porous to ensure the liquid retention and the transport of vapor. It also needs to possess high liquid entry pressure (LEP), high permeability, good thermal and chemical stability, low thermal conductivity and strong mechanical strength, narrow pore size distribution, proper thickness, etc. [36].

Over the years, efforts have been made to understand membrane characteristics and to develop novel membrane materials and modified membranes for MD. Both organic polymeric membranes and ceramic membranes are all desired membrane materials for MD. Ceramic membranes can be operated under harsh conditions, but its manufacturing cost is much higher so it is less favorable in commercial applications. Polymeric membranes are cheaper and are easier to be modified for MD applications, although they have many disadvantages, such as poor thermal and chemical stability.

### 2.2. Membrane Crystallization (MCr)

Similar to MD, MCr technology is also a promising separation technology which combines membrane separation and the crystallization process, as shown in Figure 8. In fact, MCr process can be regarded as an extension of MD process. We can also consider MD as a pre-concentration step of MCr. The feed solution is firstly concentrated by MD process, and then further concentrated until supersaturation for obtaining crystals. The MCr process can promote crystal nucleation and growth in a well-controlled way, thus adjusting the final properties of crystals, including its structure (polymorph) and morphology (crystal shape, crystal size and crystal size distribution) [37]. Furthermore, the integrated technology of MD and crystallization further improves the possibility of treating some concentrated solutions. In this way, a high-quality water can be obtained, as well as crystal products with controllable properties, so as to avoid the pollution of water resources caused by the discharge of these wastewaters to the surface or underground [38].

MCr adopts microporous membranes as the media to transfer solvents. The total porosity of the membrane dominates the transmembrane flux, and the transfer characteristics are related to the membrane pore characteristics. It includes pore size, pore size distribution and channel curvature [39]. Therefore, the membrane used in MCr not only serves as the interface for vapor transport, but also as the role for controlling the supersaturation, nucleation and crystal growth.

Concentration polarization effect is also present in MCr process. There is a boundary layer in the feed side near the membrane surface where the concentration of the nonvolatile component is higher than that in the bulk solution. The microporous structure of the membrane can also embed the solute molecules which lead to a reinforced supersaturation. When the supersaturation occurs in the boundary layer, the interaction between the membrane surface and the solutes initiates the nucleation. After the nuclei growing for a period of time, they gradually aggregate into clusters and then into crystals. Then the gradually moving solvents promote crystallization in the bulk solution. Therefore, the membrane surface serves as a physical barrier and promotes heterogeneous nucleation [40].

Configurations for MCr are actually the same as that of MD, since MCr can be seen as the further concentration of feed based on MD. That is, DCMCr, VMCr, AGMCr and SGMCr (as indicated before). Among which, the DCMCr is the mostly utilized configuration for MCr because of its simple equipment design; SGMCr is usually used for removing volatile compounds from the feed solution such as in the food industries. The principles of these configurations have been already reported in Figure 7.

Other unconventional configurations include solvent/anti-solvent crystallization. These configurations adopt a mixture of solvents and anti-solvents in a solution as the original feed solution [10]. The chemical potential gradient between the two sides of the membrane generates a driving force (such as a temperature difference or a vapor pressure difference caused by the solvent and anti-solvent concentration difference, since the vapor pressure is higher for the solution with higher concentrations of solvent). The solution with a higher amount of anti-solvent has lower solubility for the solutes, so it is easier to precipitate them from the solution. This is also related to the anti-solvent addition configuration. In this type of configuration, the solute also dissolves in the mixture of solvent and anti-solvent solution, with the other side of the membrane gradually adding anti-solvent by generating a chemical potential difference. The addition of the anti-solvent enhances the supersaturation homogeneity of the solute in the solvent and therefore the solute crystallizes [41]. Another configuration related to membrane-assisted crystallization is solid hollow fiber cooling crystallization (SHFC). The feed solution is fed into the lumen side of the membrane and the cooling solution circulates in the shell side. When the temperature of the cooling solution is lower than the feed saturation temperature, the crystallization occurs [6].

As the driving force of MCr is the trans-membrane vapor pressure difference, increasing the feed temperature can improve the mass transfer by increasing the saturated vapor pressure on the feed side. However, higher temperatures may cause a high risk of membrane fouling for those solutes with low solubility at high temperatures. Therefore, it is necessary to find the appropriate operating temperature on the feed side for the different solutes. Since the crystals’ nucleation could be formed on the membrane surface, it is easier for the membrane suffering from fouling. For the application of MCr in desalination of seawater, calcium is the common chemicals precipitated on the membrane surface. Several methods have been used in MCr to avoid this type of fouling, such as the chemical pretreatment of the feed to be concentrated via MCr, for example by adding NaHCO_3_/Na_2_CO_3_ for removing 98% of Ca^2+^ [42]. Pretreatment process is often chosen to reduce organic matter in the bulk solution to avoid wetting, because the presence of organic matter can influence crystal structure. MCr process can also be combined with other technologies to fulfill long-term operation [43].

## 3. Fluoropolymer Materials and Fluoropolymer Membranes for MD/MCr

Common hydrophobic polymeric membrane materials include polypropylene (PP), PVDF, polyvinyl chloride (PVC), PTFE, polyethylene (PE), ECTFE, PCTFE, polysulfone (PSF/PSU) and polyethersulfone (PES), etc. Among all the hydrophobic membrane materials used in MD and MCr, fluorinated polymeric membranes are popular for researchers, due to their superior chemical and physical characteristics compared to non-fluorinated polymeric membranes [44]. The most representative fluoropolymer membrane materials for MD and MCr processes are PVDF, PTFE and ECTFE (see Table 2). Among them, PVDF is the most used membrane material, and has the advantages of easy preparation, low cost and excellent comprehensive performance. It has been widely applied in a wide variety of membrane processes as ultrafiltration (UF), microfiltration (MF), pervaporation (PV), MD and MCr [45]. Although PTFE has higher fluorination degree and better hydrophobicity than PVDF, its preparation process is more complex and difficult, in fact it is not soluble in traditional organic solvents and its production cost is higher. PTFE can be prepared through extrusion, pressing, stretching and sintering under high temperature, and it has poor compatibility with other polymers and solvents [46]. These preparation methods require the use of several process parameters such as very high temperature, high stretching ratio, which make the preparation methods more complicated and costlier [47]. ECTFE is a membrane material that has been utilized in MD in recent years. It has high hydrophobicity and high separation performance, being thus a promising candidate for MD applications as well as for MCr applications.

### 3.1. Poly(Vinylidene Fluoride) (PVDF)

#### 3.1.1. PVDF and Its Copolymers

PVDF homopolymer and some of its main copolymers are membrane materials that have been applied in MD process due to their chemical and thermal properties combined with easy fabrication, related to the use of phase inversion techniques. They have many advantages with respect to other polymers, such as flexibility, non-toxicity, easy processability, outstanding dielectric permittivity, etc. They are an important class of electro-active smart materials that show pyroelectric, piezoelectric and ferroelectric properties [49]. PVDF homopolymer has good chemical, mechanical and heat resistance properties. It is a semi-crystalline polymer containing a crystalline phase and an amorphous and/or rubbery phase [50]. It has five kinds of phases—α, β, γ, δ and ε (Figure 9).

These five different phases can be distinguished through Fourier transform infrared spectrometry (FTIR) and X-ray diffractometry (XRD) analysis [52]. The α-phase follows a (TGTG’) (T denotes the trans, G denotes the gauche) atomic arrangement in a unit cell. The CH_2_ dipoles are perpendicular to CF_2_ units, so it produces an electric dipole that is perpendicular to the axis with corresponding piezoelectric and ferroelectric charges. It is reported that the TGTG’ conformation is the most stable one. It is obtained when the defect concentration in the membrane is higher than 11 mol%. It is the only electrically inactive non-polar phase. The β-phase has the crystal structure with CH_2_-CF_2_ dipoles oriented in the same direction and has the (TTTT) planar zigzag conformation [53]. It is normally obtained from the melted process membrane with a defect of lower than 11 mol%. To increase the piezoelectricity of the membrane, or conversion of the mechanical energy into electrical, or vice versa, it is necessary to increase the β-phase content of the polymer. Many researches have been applied in this direction by making nano-forms, composites by electrospinning, 3D printing, using multilayer materials such as incorporation of graphene, carbon nanotubes, metallic nanoparticles, etc. Other methods include blending the base polymer with other polymers containing carbonyl groups, which is due to the fact that the interactions of the dipole–dipole between carbonyl and polar fluorination groups. This method can be directed into the incorporation of the piezoelectricity properties [54]. The γ-phase has the (T_3_GT_3_G’) conformations and the CH_2_-CF_2_ dipoles are parallel to each other to form a non-centro-symmetric polar crystal. δ and ε-phases are similar to that of α and γ-phases, respectively. These different phases can be transformed and under various processing such as heat treatment, stretching and electrical application.

Common PVDF copolymers include poly(vinylidene fluoride-co-tetrafluoroethylene) (P(VDF-co-TFE)), poly(vinylidene fluoride-co-hexafluoropropene) (P(VDF-co-HFP)), poly(vinylidene fluoride-co-chlorotrifluoroethylene) (P(VDF-co-CTFE)), PVDF-g-PSSA, and poly(vinylidene fluoride-trifluoroethylene) (P(VDF-co-TrFE)), etc. (see their structures in Figure 10 [55]). The former three copolymers have been reported in MD application with flat sheet and hollow fiber membranes prepared by phase inversion (PI) technique [56,57,58].

P(VDF-co-HFP) copolymer has relatively lower crystallization temperature (T_c_) and higher F compositions compared with PVDF homopolymer, which will result in higher hydrophobicity [59]. The intrusion of HFP groups in the PVDF polymer also enhances its solubility, mechanical strength, and lowers the crystallinity [60]. Therefore, it is also promising in its application as polymer electrolytes, such as in lithium batteries, due to its lower crystallinity. It is normally blended with some inorganic nanoparticles such as TiO_2_ [61], graphene oxide (GO) [62], SiO_2_ [63], Co-ZnO [64], etc. to improve its conductivity properties in electrolytes. This could help to produce high performance nanocomposites materials and make them adapt for the energy harvesting and storage applications [65].

P(VDF-co-CTFE) is also a semi-crystalline polymer with lower crystallinity degree (around 5.5–12.6) compared with that of PVDF (50–70%) [66]. It has high flexibility, high electromechanical response, high elongation and cold resistance compared with other PVDF copolymers. It exhibits higher piezoelectric constant (d_33_) than that of PVDF homopolymer. A small amount of CTFE is incorporated into the polymer, it shows a gauche kink to stabilize the TGTG’ conformation to avoid phase transition under a high electric field. The elements of Cl and F in P(VDF-co-CTFE) can enhance the safety of the separators due to the fact that they have flame-retarding functions [67]. The presence of CTFE chain segment in the molecular chain can not only enhance the mechanical properties of the polymer, but also enhances the thermal stability and chemical corrosion resistance. The existence of a C-Cl bond (327 KJ mol^−1^) can present an active site for modification since its bond energy is lower than the C-F bond (486 KJ mol^−1^) [68]. Therefore, P(VDF-co-CTFE) has great potential as a membrane material in the application of a membrane field [69]. The chemical structures of PVDF and their copolymers are shown in the following Table 3.

P(VDF-co-TFE) is a polymer based on the copolymerization of PVDF and TFE. It is less investigated than P(VDF-co-TrFE). TFE is the fluoro monomer utilized to fabricate PTFE membrane. It is also regarded to have stronger capability than the TrFE monomer in terms of the trans-bond stability in the copolymer. It can be dissolved in common solvents such as DMSO, DMAC, and NMP, etc., therefore generating strong hydrophobicity.

#### 3.1.2. PVDF and Its Copolymer Membranes

Commercial PVDF membranes are available from Merck Millipore (Burlington, MA, USA), Solef Solvay, Arkema (Beijing, China), Aquastill (Sittard, The Netherlands), GE Healthcare, DOW company, Suntar company (Xiamen, China), etc. They can be processed into flat sheet, hollow fiber and tubular morphologies. PVDF membranes can be fabricated by phase inversion (PI) process, thanks to its solubility in common organic solvents [88]. Many studies have been conducted on the preparation of superhydrophobic PVDF membrane and their applications in various different processes [89]. PVDF membranes also have five kinds of phases as their starting polymer. They show different properties with different phases, such as an anti-fouling property or a piezo-electric property [90]. Cao et al. [51] prepared β-phase PVDF flat sheet membrane by using ionic liquid [BMIM]PF_6_ via TIPS method. The membrane endowed piezoelectric properties when applied under high voltage electricity. It is one of the most investigated membranes, that has been used in hydrophobic membrane preparation and applied in the MD process. Examples of fluoropolymer membranes that have applied in MD and MCr in recent years are shown in the following Table 4. Among all the fluoropolymer membranes, PVDF, PTFE, ECTFE, P(VDF-co-HFP), P(VDF-co-TFE) and P(VDF-co-CTFE) membranes, etc., were applied in MD until now. While for MCr, only PVDF, PTFE and P(VDF-co-HFP) membranes were applied in this process. In principle, membranes used for MD have the potential to be used also in MCr, while the membranes used for MCr can definitely be used in MD. Since the feed concentration used for MCr is near the saturation level, the membrane properties used for MCr should have a higher resistance toward the high concentration of the feed solution.

The SEM images of different morphologies of PVDF membranes are shown in the following Figure 11. The prepared PVDF membranes generally show porous, finger-like, sphere-like and ultrafine morphology under SEM. They are about the surface and cross-section of PVDF membranes prepared using different membrane techniques. Among these techniques, the ultrafine membrane was prepared by the electrospinning method and the other three types of membranes were prepared by phase inversion methods. As known, the preparation methods have an impact on the final membrane morphology. In principle, membranes used for MD and MCr should be hydrophobic and porous, therefore membranes with these four types of surfaces can all be used for MD as long as they are hydrophobic. In general, the asymmetric structure with a thin, porous layer supported by finger-like structure should give higher fluxes due to lower resistance to vapor transportation; however, also the overall porosity dominates the flux for both MD and MCr processes.

PVDF copolymer membranes can also be utilized in MD and show excellent performances [106,107]. P(VDF-co-HFP) membranes exhibit better performance than that of PVDF membranes. However, they have a low piezoelectric voltage constant so that they need a high electric field for polarization, making polarization more difficult. Studies [108] were reported on the P(VDF-co-HFP) blended with other polymers such as poly(methyl methacrylate) (PMMA), PVC, poly(vinyl acetate) (PVAc) for electrolytes applications. The preparation of the composite P(VDF-co-HFP) membranes grafted by cross-linked collagen was also reported. Glutaraldehyde (GLA) was used as a covalent crosslinking agent of collagen. The crosslinkers were used to enhance the insolubility and mechanical strength of the polymeric membrane. GLA reduces membrane swelling and also improves the mechanical strength of the composite membranes. CNFs were used as fillers in this study, and GLA-PVDF-HFP/CNF composite membranes were finally prepared and applied in DCMD process. In addition, GLA is also reported for the preparation of high performance in high flux, fouling resistance and long-time operation in MD. The effect of PEG additive was investigated on the P(VDF-co-HFP) membrane porosity, surface morphology, thermal stability, hydrophobicity and VMD application. The operation in VMD showed a 100 h stability in terms of flux and with rejections above 99.8% [109].

P(VDF-co-CTFE) membranes can also be fabricated by NIPS [110], TIPS and electrospinning methods [111]. Specifically, Lee [112] et al. prepared P(VDF-co-CTFE) membranes with enhanced fouling (almost no flux decline during the DCMD) and wetting resistance for improving their performance in DCMD. The prepared membrane was functionalized by 1H, 1H-perfluorooctyl methacrylate (FOMA) to improve its surface hydrophobicity and long-term MD tests. The results show that the membrane has super-hydrophobicity and excellent anti-fouling and wetting properties.

P(VDF-co-TFE) membrane has better mechanical strength than PVDF membrane. As indicated before, P(VDF-co-TFE) membranes have strong hydrophobicity and they can be fabricated by phase inversion method. Feng et al. [57] prepared P(VDF-co-TFE) membrane by dissolving the polymer into DMAC at 50 °C, LiClO_4_·3H_2_O or/and TMP were utilized as additives and pore formers. The membrane was fabricated by NIPS method and tested in MD. The performance of the membrane was compared with that of the PVDF membrane. Results showed that the contact angle of the prepared membrane increased and DCMD tests showed that the flux increased to 7.3 kg/m^2^h, which was higher than that of the original PVDF membrane (6.7 kg/m^2^h) when feed temperature was at 55 °C and permeate side temperature was 20 °C.

### 3.2. Poly(Tetrafluoroethylene) (PTFE)

#### 3.2.1. PTFE and Its Copolymer

PTFE was discovered by Dr. Roy J. Plunkett at the DuPont Company, and it is known as Teflon. It is produced by polymerization of the monomer TFE and the molecular formula is [(CF_2_-CF_2_)_n_] [80]. Bunn and Howells first reported its crystalline structure in 1954 [113]. PTFE has a similar crystalline structure of polyethylene (PE) but does not have a zigzag conformation, which crystallizes in a zigzag shape only at extreme pressure, at 5000 atm. It is well known for its tribological applications. PTFE’s melting point is at around 325 to 335 °C, so it is defined as a thermoplastic due to its high operating temperature and thermal resistance. Owing to its superior advantages, such as chemical resistance, electro-physical properties, and optical characteristics, it attracts much attention in the engineering field, including as a membrane [114]. It is also a semi-crystalline polymer, and its linear chains show complex phases in crystalline domains (Figure 12). Its crystalline structure transformation occurred at 19 °C from triclinic structure to hexagonal structure and to pseudohexagonal structure at 30 °C [115]. PTFE is insoluble in most solvents and is also resistant to almost all acidic and caustic materials [79]. Because of its low surface energy, it almost has no adhesion to other materials; it also has a limited resistance to heat. These disadvantages limit the application of this material. It is normally classified as additive [116] or filled with micro/nanoparticles to fabricate composites to improve its properties. PTFE homopolymer and its copolymers also have potential in membrane preparations. PTFE copolymers include perfluorosulfonic acid (PFSA), poly(tetrafluoroethylene-co-perfluoropropyl vinyl ether) (PFA), poly(tetrafluoroethylene-co-hexafluoropropylene) (FEP), and poly(ethylene-alt-tetrafluoroethylene) (ETFE), etc. Among these, FEP is used in MD.

FEP is a copolymer of tetrafluoroethylene and hexafluoropropylene. It is different from PTFE polymer, and it can be produced by using conventional injection molding and screw extrusion techniques. It is sold under the brand name Teflon FEP by the Dupont Company. It is also named Neoflon FEP and Dyneon FEP by the 3M Dyneon company. Its composition is very similar to that of the PTFE fluoropolymer but is more easily formable. Its melting temperature is at around 260 °C (500 °F) and it is transparent and sunlight resistant. It is a pure carbon–fluorine structure that is fully fluorinated, which makes it the only fluoropolymer that has comparable corrosion resistance to PTFE. Therefore, it is a promising candidate for replacing PTFE polymer and extending its application in different membrane fields.

#### 3.2.2. PTFE and Its Copolymer Membranes

As mentioned above, PTFE membranes show many superior advantages, which make them suitable in many applications such as oil–water separation, membrane contactors (MD and OD), etc. Porous PTFE membranes are normally fabricated by different techniques such as spinning, paste extrusion, stretching and/or sintering, pore-forming, etc., which are shown in Section 4.5.1 [118]. In desalination, asymmetric PTFE membranes with high separation ability are used in RO for separation of contaminants, sediments, and salts due to their good durability and strength [80]. In MD application, the pore size used in PTFE membranes is usually around 0.5 μm. The surface energy of PTFE membrane is 9.1 kN/m and its thermal conductivity is as low as 0.22–0.45 Wm^−1^K^−1^, which make it desirable for MD. The porosity of the PTFE membrane can be as high as 90%. It is also not as rigid as a PVDF membrane that can deform under high pressure [32].

Commercial PTFE membranes include flat sheet and hollow fiber membranes. Some commercially available PTFE membrane products are Teflon (DuPont), Polyflon (Daikin), Dyneon PTFE (Dyneon), and Fluon (Ashai Glass). Modifications of the PTFE membrane surface were also applied to improve their hydrophobicity, anti-fouling resistance, or to functionalize the membrane with other properties. These methods include the wet-chemical method, plasma treatment, irradiation, atomic-layer deposition, high temperature melting, etc. Each method is chosen based on the surface properties of the membrane and the requirements of the applications.

For MD applications, Chen et al. [119] investigated a PTFE flat sheet membrane in DCMD. It was laminated by two layers of PTFE membrane to increase the duration of MD process in highly concentrated NaCl solution. Hollow fiber PTFE membranes are also commonly used in MD [120]. Ultrafine fibrous porous PTFE composite membranes with PVA as additive by electrospun sintering were also fabricated by Huang et al. [121]. The SEM images of these membrane morphologies are shown in Figure 13, which shows that through different preparation methods of the membrane, the surface morphologies of the membrane are different, being useful for targeting specific applications. These three types of membranes were all applied in MD. As already indicated in the PVDF section, the membranes used in MD have potential to be applied in MCr applications, which is also suitable for PTFE membranes. At the moment, although hollow fiber PTFE membranes have already been applied in MCr, the study of PTFE membrane in MD process is still limited due to its difficult fabrication process and high cost.

FEP membranes show excellent properties comparable to PTFE membranes. It is a promising fluoropolymer membrane material that has already been applied in oil–water separation, MD, fuel cell [123] and energy saving fields [124]. Huang et al. [125] fabricated PTFE-co-FEP ultrafine membranes by electrospinning and sintering method. The prepared membrane showed tubular structure, with superhydrophobic and superoleophobicity properties and excellent selectivity and flexibility. Chen et al. [126] prepared FEP hollow fiber membrane by the melt spinning method. The different stretching ratios on the membrane structures were investigated. The prepared membranes were applied in VMD by using alkaline solution as feed under high temperature. Results show that the rejection rate of the FEP hollow fiber membranes exceeded 99.0%. The flux and mechanical strength remained stable during the long term alkaline aqueous MD operation. The addition of DOP and SiO_2_ to FEP flat sheet membrane improved the performance of VMD. The salt rejections increased as high as 99.9% and the flux was 2.6 LMH [127].

### 3.3. Poly(Ethylene Chlorotrifluoroethylene) (ECTFE)

#### 3.3.1. ECTFE and Its Copolymer

Poly (ethylene chlorotrifluoroethylene) (ECTFE), known with the trade name of Halar^®^, is a copolymer of ethylene and chlorotrifluoroethylene with the repeated unit -(-CH_2_-CH_2_-CFCl-CF_2_-)_n_-. ECTFE is prepared by the copolymerization of ethylene and CTFE at lower temperature (<10 °C) by using a catalyst in an aqueous medium. Its polymer chain adopts a zigzag confirmation. It has a melting temperature at a range of 220–245 °C. ECTFE has superior properties especially for the high hydrophobicity, wear resistance, chemical stability than other commonly used membrane materials. It is a promising candidate for MD. ECTFE is insoluble in most solvents at ambient temperature, but soluble in some solvents at high temperatures, so membranes can be prepared by TIPS method. The TIPS method has many advantages over the NIPS method such as easy control, higher porosity, more flexibility, and controllable pore size during the membrane fabrication process. However, recently Solvay Specialty Polymers developed a low melting point (LMP) ECTFE, which has comparable properties with standard Halar^®^, but has lower crystallinity and a lower melting point [97], which makes it soluble at lower temperature in several solvents.

#### 3.3.2. ECTFE and Its Copolymer Membranes

ECTFE membranes have been produced in recent years. It has stronger acid-base resistance, oxidation resistance and corrosion resistance than PVDF and other membrane materials. ECTFE is the most abrasion resistant and has the highest tensile strength among all the fluoropolymer membranes available at the moment [26]. Examples of ECTFE porous membranes (SEM images) that applied in MD are shown in Figure 14. A green solvent–acetyl tributyl citrate (ATBC) was also introduced into the membrane preparation procedure, the polymer concentration and quenching temperature on the membrane properties were investigated [128]. From the SEM images, it can be seen that the spherulite structure was formed and the test in VMD also presented good performance (the permeate flux reached to 22.3 L/(m^2^·h)). Other types of ECTFE membranes (surface and cross-section of hollow fibers) are also shown in the following Figure 14. This membrane was prepared by TIPS method using also green solvents (DEP or GTA). The bicontinuous structures were observed from SEM images. An improved mechanical strength and water permeability was obtained compared to previous ECTFE membranes either with hollow fiber or flat sheet membranes.

Ursino et al. [97] prepared non-woven supported ECTFE membranes, by the dip-coating method, which were successfully applied in DCMD. DEA and DEG were used as primary and secondary diluents. DEG also plays the role of pore former, and the effect of immersion time was investigated. Pore size, porosity and contact angle were characterized for the prepared membranes. The prepared membranes were applied in DCMD by using pure water and 0.6 M NaCl as feed. Fluxes were ranging from 3–22 LMH, depending on the different DEG concentration, immersion time and feed temperature (40–60 °C), and the salt rejections were ranging from 94.95% to 99.82%.

### 3.4. Other Fluoropolymer Materials

Apart from the mentioned fluoropolymer materials, other fluoropolymers can be potentially used in membrane preparation, such as poly(chlorotrifluoroethylene) (PCTFE), PVDF-co-TrFE, Hyflon^®^ AD, Teflon^®^ AF, Cytop^®^, PVF, perfluoropolyether (PFPE), etc. These fluoropolymers have high hydrophobicity and some of them have already been used as coating materials on the membrane surface. Some fluoropolymer materials have not been applied in MD but have great potential to be used in MD and MCr.

Hyflon AD, is a kind of copolymer of TFE and 2, 2, 4-trifluoro-5-trifluoromethoxy-1, 3-dioxole (TTD). It has drawn great attention due to its excellent hydrophobicity, high temperature stability and chemical resistances. It can form uniform and thin layer of membranes by casting or coating methods [129]. A coating of Hyflon AD 40 and Hyflon AD 60 on original PVDF membrane to form composite membranes has been tested recently [130]. The results showed that the hydrophobicity increased, and the mechanical strength was also enhanced. The final composite membranes worked for a long time during MD operations. Additionally, the wetting and fouling resistances were enhanced [130]. Actually, these materials can be used alone to prepare the membranes due to their extremely high hydrophobicity and anti-fouling resistance. However, their use is limited due to the high cost (such as Hyflon AD 60 at around 200€/g). This is also the reason why these materials are normally used as coating materials to modify the surface of the membranes to improve their hydrophobicity.

Teflon^®^ AF is a kind of copolymer that has desirable chemical, thermal and electrical properties. It is a family that consists of tetrafluoroethylene (TFE) and 2,2-bis-trifluoromethyl-4,5-difluoro-1,3-dioxole, (PDD). Their properties vary with the concentration of PDD in the polymers. Teflon^®^ AF-1600 and Teflon^®^ AF-2400 are two commercial polymers that contain 64% and 83% PDD, respectively. Lu et al. [131] prepared PVDF membranes by using Teflon AF 2400 as coating material to improve its hydrophobicity. The membranes were tested in VMD with sodium dodecyl sulfate (SDS) as feed. They were also coated on tri-bore hollow fiber membranes and applied in DCMD [132]. A super hydrophobic surface of 151° was obtained, and the membrane showed an average flux of 22 L/m^2^h and salt rejection of 99.99% with a feed temperature of about 60 °C.

PFPEs have the characteristics of high thermal and chemical stabilities, which make them highly attractive under specific or extreme conditions. It is also soluble in most solvents under high and ambient temperatures. Figoli et al. [133] investigated PFPE oligomer compounds (Fluorolink^®^ AD 1700) for preparing hydrophobic/hydrophilic coated membranes. The prepared membranes were highly hydrophobic under UV curing. A flux of 11 kg/m^2^h and rejection of 99.3% were obtained for 0.6 M NaCl solution when feed temperature was 50 °C, indicating the good performance of the membrane in DCMD and the stability of the coating. The biggest advantage of PFPE material is that it can be coated on the cheaper hydrophilic commercial membranes (such as PA) to modify their surface, which is more economical than the normally utilized expensive membrane materials such as PTFE, PVDF, PP and ECTFE. Coating of PFPE on PVDF membrane was performed to improve its anti-wetting and anti-fouling properties. The contact angle increased to 162°, and the application in VMD in 2 M NaCl solution with 200 mg/L humic acid (HA) showed more than 99.99% salt rejection and the recovery of flux was 90% of the initial flux [94].

P(VDF-co-TrFE) is a copolymer based on the combination of VDF and TrFE. It can be directly obtained by the traditional copolymerization method of VDF and TrFE in dispersed suspension. Other methods include reductive dichlorination, such as the reduction of the chlorine from the P(VDF-co-CTFE) copolymer. Based on different preparation methods of the polymer, it displays different chain conformations, crystalline structures, and also different polarization behaviors. Traditional P(VDF-co-TrFE) contains 50–80% VDF, obtained by melting and casting which shows trans-conformation without extra stretching operation. Due to the incorporation of TrFE units, the polymer has a larger inter-chain distance (d_110/200_ = 0.442 nm) compared with that of β phase PVDF (d_110/200_ = 0.426 nm). This makes it easier for rotation around the chain axis. It displays lower crystallization temperature (T_c_) than melting temperature (T_m_), which could experience the transition from ferroelectric to paraelectric phases. T_c_ of the P(VDF-TrFE) copolymer varies from 60 to 135 °C as the molar fraction of TrFE increases from 20 to 50% [71]. P(VDF-co-TrFE) membrane is attractive due to its high piezoelectric activity, easy to be fabricated into fibers and membranes, flexibility, etc. Its piezoelectric properties are mainly from β phase. To obtain β phase, one of the most effective ways is through electrospinning in the high electric field [134]. Actually, to the best of our knowledge, there is no reported research for this type of membrane applied in MD. However, it has great potential in MD application, which needs further exploration [65,66].

Polychlorotrifluoroethylene (PCTFE), a fluoropolymer that has similar properties to PVDF, but is less deformable and less gas permeable, is commonly utilized in gas separation but might be introduced also in MD or MCr. It has a melting temperature of 190–224 °C and its crystallization temperature is around 150–190 °C. The molecular weight of PCTFE is around 74–510 kg/mol. The crystallinity of PCTFE is at around 0.3–0.55. Pores of the PCTFE membrane are so small that it needs adding pore formers to increase its pore size to the UF/MF range to be suitable for MD [83].

PVF is normally prepared by polymerization of vinyl fluoride (VF) in the aqueous medium under higher pressure compared to TFE. It is prepared by using melt extrusion with some stabilizers and plasticizers inside. Its molecular weight is around 228–790 kg/mol. The melting temperature of PVF is around 178–200 °C. Its crystallization temperature is around 130–180 °C. The crystallinity of PCTFE is approximately 0.12–0.68 [84,85,86]. It can also be utilized in temperatures around 70 °C to 110 °C. The crystallinity of PVF is lower at around 40% compared with that of PTFE, which will lead to a higher tensile strength. PVF membranes have been used as solar energy collector materials due to their good wear resistance and anti-radiation properties. It is currently produced by the DuPont company [26]. It can be blended with other membrane materials to improve its performance. It has great potential in MD applications.

## 4. Preparation of Fluoropolymer Porous Membrane for MD/MCr

### 4.1. Preparation of PVDF Membrane

#### 4.1.1. Non-Solvent-Induced Phase Separation Method (NIPS)

The NIPS method is the most commonly used method to prepare porous UF/MF membrane at an industry level. The membrane preparation process and phase separation principle are shown in Figure 15. The basic principle is that the polymer solution is scraped onto the support or extruded and then immersed in a non-solvent bath (NIPS technique). In the bath, when the solvent in the polymer solution is dispersed into the non-solvent, the non-solvent also diffuses into the polymer solution, and therefore the phase separation is formed under the bi-directional dynamic diffusion. Finally, the polymer gels and precipitates in solid form [26]. For NIPS method, the phase composition of the system changes during phase separation, which ultimately determines the membrane pore structure. Phase separation is the basis of pore formation.

Figure 16 is a typical ternary phase diagram of a polymer/solvent/non-solvent system. The mixed system outside the binodal curve (region 1) is in the thermodynamically stable region that forms a stable starting polymer solution, then there is a thermodynamic metastable region between the binodal curve and the spinodal curve (region 3). According to whether the system passes through the critical point or not, the liquid–liquid phase separation process can be divided into: the spinodal phase separation (passing through the critical point) and the nucleation phase separation (without passing through the critical point). Generally, if the composition of the system moves from below the critical point into the region 3, the system obtains a spherical structure with low mechanical properties; if the system enters the region 3 from above the critical point, the bi-continuous structure is obtained.

For PVDF membranes utilized in MD, the NIPS method is the most used one. There are many factors influencing the NIPS membrane preparation process. For example, the types of PVDF polymer and their initial concentration [36], the type of solvent and non-solvent [135], the composition and temperature of casting solution [136], the composition and temperature of coagulation bath [137], the characteristics of PVDF gelation and crystallization, and evaporation time [138,139] have been investigated for affecting the final structure and properties of the PVDF membrane. These properties also affect the membrane performance in MD. It is reported that the molecular weight of the PVDF polymer also affects the viscosity of the dope solution, which will finally affect the membrane surface energy and roughness. Chen et al. [140] compared three types of PVDF polymers with different molecular weight and applied them in VMD. Results show that the membrane surface roughness, surface energy and porosity all increased with increasing molecular weight, while the contact angle decreased with increasing molecular weight.

#### 4.1.2. Thermally-Induced Phase Separation Method (TIPS)

Thermally-induced phase separation (TIPS) is a method for preparation of microporous polymeric membranes proposed by Castro [141] in 1981. The polymer is mixed with a specific diluent at high temperature to make a homogeneous casting solution, and then phase separation occurs during the cooling process. After preparation of the membrane, the corresponding extractant is selected to extract the diluent, and finally the microporous polymeric membrane is prepared. In short, it is “Dissolution under high temperature, and phase separation under low temperature”. The preparation of polymeric membrane by TIPS is mainly affected by the concentration, cooling rate, quenching temperature, diluent and additives. Temperature is the driving force of the whole phase separation process. The quenching temperature and cooling rate determine the pore size and the spherulite size, and ultimately affect the overall separation performance and mechanical properties. Membranes prepared by TIPS method usually have narrow pore size distribution, high porosity and the easy controlled microstructure [142]. The TIPS method was originally used to solve the problem of polymer materials (such as polyethylene, polypropylene, etc.) that is not suitable for the NIPS process, but it is widely used also for PVDF. Compared with the PVDF membrane prepared by NIPS, the strength of PVDF membrane prepared by TIPS has been greatly improved, thanks to the possibility of using a higher concentration of polymer in the starting dope solution, and the tensile strength can reach more than 10 MPa [143].

The common phase diagram of the TIPS method is shown in Figure 17. The liquid–liquid phase separation region and the solid–liquid phase separation region are separated by the dynamic crystallization line. The intersection point of the binodal curve and dynamic crystallization line is called the monotectic point, of which the corresponding polymer concentration is the critical concentration to distinguish the liquid–liquid phase separation and the solid–liquid phase separation in the system. For cooling paths 1, 2 and 3, the corresponding polymer concentration is lower than the monotectic point concentration, therefore resulting in liquid–liquid phase separation, and the obtained membrane is with bi-continuous structure. However, for path 4, the polymer concentration is higher than the monotectic point, thus resulting in solid–liquid phase separation, and the solidified membrane shows obvious spherical accumulation structure, indicating that liquid–liquid phase separation does not occur [144]. Generally speaking, when the polymer content in the casting liquid system increases gradually, the structure of the prepared membrane becomes more compact, and the porosity and pore size of the membrane will be smaller [145].

For the membrane preparation process of TIPS, solutions with good solubility and suitable interaction with polymers are generally selected as diluents, among which phthalic diluents such as dimethyl phthalate (DMP), diethyl phthalate (DEP), dioctyl phthalate (DOP) and dibutyl phthalate (DBP) [146] are representative, but these diluents are volatile at high temperature, therefore more and more researchers have begun to study environmentally-friendly diluents and multi-diluents. PVDF, ECTFE flat sheet membranes and hollow fiber membranes have then been prepared by using environmentally-friendly diluent tri-n-butyl acetylcitrate (ATBC) [128,147]. The experimental results show that the increase of polymer concentration and quenching temperature is conducive to the densification of membrane structure, improving the mechanical properties, and reducing the pore size and porosity of the membrane. The contact angle of ECTFE membrane can reach 140°, the flux can reach a value of 22.3 L/(m^2^·h), and VMD process has a salt rejection of 99.9%. ECTFE flat sheet membrane by TIPS method was also prepared with trioctyl trimellitate (TOTM) as green diluent. The prepared membranes also presented high hydrophobic properties, excellent permeability and high salt rejection in VMD process [148].

#### 4.1.3. Vapor-Induced Phase Separation Method (VIPS)

Vapor-induced phase separation (VIPS) was first proposed in 1918. It is a kind of membrane preparation method that first put the original membrane in the non-solvent vapor environment for a period of time, and then immerses it in the non-solvent coagulation bath to solidify the polymer into a membrane. For the membrane preparation process of VIPS, the main reason for phase separation is the inflow of non-solvent, not the outflow of solvent (as shown in Figure 18). Compared with NIPS method, the phase separation in VIPS process is more stable, and the mass transfer rate (non-solvent absorption and solvent extraction) is greatly reduced, which can effectively avoid large pore defects. During the membrane preparation process, the non-solvent phase is a gas; for the technical problems of controlling the gas state, at present, only a few gases are reported to be used in VIPS process. The most common one is humid air, and in addition, there are water vapor, acetone vapor and ammonia vapor. Because the phase separation of VIPS mainly occurs at the environment via a non-solvent, the requirement for environmental conditions is high; the temperature and humidity all have a crucial influence on the final membrane morphology. Therefore, the membrane preparation process of VIPS is generally completed in glove boxes or closed thermostatic glass chambers to control ambient temperature and humidity conditions [149,150].

VIPS method can be used to prepare polymeric membrane with a rough surface, so that it has high hydrophobic properties. Xie [151] and Mao [152] prepared porous and highly hydrophobic PVDF membranes with micro and nano-graded roughness by the VIPS method. Fan [153] used 100 RH% high humidity gas to prepare a hydrophobic PVDF membrane by the VIPS method. With the increase of vapor induction time, the membrane pores changed from asymmetric finger-like pores to symmetric sponge-like pores, and the membrane surface became rough and porous. The contact angle of hydrophobic membrane prepared by long time vapor induction could reach 145°. The membrane prepared under higher vapor-induced time also present higher flux in VMD. The best membrane showed a flux of 22.4 LMH at the feed temperature of 73 °C and NaCl rejection of 99.9% for 3.5 wt% concentration feed solution. The VIPS method can also be combined with NIPS for fabricating PVDF membranes. Russo et al. [154] prepared a PVDF membrane by V/NIPS method using TEP as a non-toxic solvent. The obtained membrane showed the asymmetric structure and the pore size in the range of UF membrane when a low concentration of PEG additive was used. The membrane showed bi-continuous structure and the humidity exposure time plays, and the important role of pore structure. The membrane showed thicker, higher porosity and more hydrophobicity when exposure time increased. The best membrane was prepared when the dope solution containing 15% PEG had an exposure time of 2.5 min before entering into a coagulation bath. The membrane presented 82.8% porosity, 0.43 μm pore size, and PWP at around 7900 L/m^2^·h·bar, which has the potential to be used for MF application. They also prepared the membrane in the range of UF that has potential to be used for MD [155].

#### 4.1.4. Evaporation-Induced Phase Separation Method (EIPS)

For evaporation induced phase separation method (EIPS), demonstrated in the following Figure 19, a homogeneous solution is prepared by dissoving a polymer in the mixture of a solvent and a non-solvent, where the solvent has higher volatility than the non-solvent. By the evaporation of the solvent, phase separation phenomenon occurs from the solvent-non-solvent solution, forming a porous membrane. The pore structures can be controlled by adjusting the constitution of solvent-non-solvent solution. The major difference of EIPS method with VIPS method is that EIPS method is the outflow of solvent, not the inflow of non-solvent. 

In general, the membrane prepared by VIPS or EIPS method has packed interconnected structure due to the liquid-solid mixing by crystallization and polymer gelation [98].

#### 4.1.5. Electrospinning

Zeleny conducted the electrospinning technique in 1914 [156]. From 1934 to 1944, a series of patents were proposed by Anton of electrospun polymer filaments [157,158]. Due to the increasing interest of nanomaterials and nanotechnologies in recent years, electrospinning is attracting more and more attention. It can fabricate nanofiber membranes with the diameters from micron to nanoscale. These membranes have the characteristics of high porosity, excellent pore interconnectivity, low density, large surface area, micron interstitial space and tailorable membrane thickness. Traditional electrospinning is composited of a high degree of voltage, one or more grounded collectors and a spinneret. It can be classified into vertical and horizontal apparatus at ambient temperature according to its position and direction. According to the solution status, it can be classified into solution electrospinning and melt electrospinning [159]. A typical set-up is shown in the following Figure 20. Firstly, the polymer solution is introduced into the syringe, a high voltage is applied between the spinneret and the collector. When the voltage overcomes the surface tension of the polymer fluid, the electric field causes the droplet to deform into a conical structure. This charged polymer solution is therefore ejected into the collector, and in the meantime, most of the solvents evaporate due to the unstable whipping motion and the continuous elongation of electrostatic repulsion between the nozzle and the collector. Finally, solution solidification makes the electrospun membranes [160,161,162].

Liao et al. [163] utilized the electrospinning method for preparing PVDF nanofiber membranes. Polymer concentration and the spinning parameters were investigated; the prepared membranes were characterized and applied in the DCMD process. Results show that the optimized membrane exhibits a rough surface of high hydrophobicity with the contact angle higher than 135° and a stable flux of 21 kg/m^2^h with the 3.5 wt% NaCl feed solution. This result was better than a commercial PVDF membrane and nanofiber PVDF-clay membranes for DCMD applications. Yao et al. [164] investigated the effect of heat-press temperature, pressure and duration on the morphology and mechanical characteristics of PVDF-co-HFP membrane and the application in DCMD. The final optimal conditions were heat-press temperature at 150 °C, pressure of 6.5 kPa, and a duration for 8 h. The DCMD flux of 29 LMH and 99.99% salt rejection were achieved at feed and permeate temperature of 60 and 20 °C.

### 4.2. Preparation of PVDF-co-HFP Membrane

#### 4.2.1. NIPS

The NIPS method could also be used for fabricating PVDF-co-HFP membranes. Fadhil et al. [165] fabricated a flat sheet PVDF-co-HFP membrane by dissolving the polymer into a green solvent TEP via NIPS method. The membrane showed a sponge-like pore structure. A series of polymer concentrations were investigated, and the lithium chloride was added as an additive to increase the pore size and porosity of the membrane. An isopropanol–water mixture was used as a coagulation bath and the membrane produced at a concentration of 12 wt.% showed good performance in a DCMD test. A flux of 16.1 kg/m^2^h was achieved under the feed temperature of 60 °C and a rejection of 99.3% was obtained.

#### 4.2.2. TIPS

Yadav et al. [102] prepared PVDF-co-HFP flat sheet membranes by incorporating CNT^@^MOF5 as additives by TIPS method. The composite membranes were characterized by SEM, thermogravimetric analysis (TGA), X-ray diffraction (XRD) and Fourier transform infrared (FT-IR) analysis. The membranes with incorporation of the nanoparticles show rougher surface and higher porosity than the original membrane. The application in MD process also shows higher flux, with 14.40 kg/m^2^h for the hybrid membrane. It was also used in recovering CaSO_4_ and NaCl from sub-soil brine by VMCr. The recovered salts show higher purity with the confirmation from SEM and XRD. Khayet et al. [166] also prepared the PVDF-co-HFP flat sheet membranes by TIPS method. Optimized preparation conditions for preparing the hybrid membranes were investigated, such as the polymer and additive concentrations, solvent evaporation time and temperature of the coagulation bath. They were also optimized for the DCMD test. The results show that under the optimized conditions of 19.1 wt.% PVDF-co-HFP concentration with 4.99 wt.% PEG, at the temperature of 35 °C coagulation bath and 102 s solvent evaporation time, that the membranes show the best performance in DCMD with the 99.95% salt rejection and 4.41 L/m^2^h flux.

#### 4.2.3. Electrospinning

Hou et al. [98] prepared superhydrophobic PVDF-co-HFP/SiNPs hybrid flat sheet membranes by electrospinning method. A polyester non-woven fabric was used as the support layer. All membranes present interconnected open morphology. The addition of the nanoparticles SiNPs enhanced the hydrophobicity of the membrane surface and the water contact angle reached more than 150°. The membrane surface presented the hierarchical structure due to the beads-on-string in the nanofibers. The membrane was tested in DCMD with 35 g/L NaCl as feed. The test was run for 240 h and the NaCl rejection remained 99.99%. The highest flux obtained was 48.6 kg/m^2^h. All these performances show great potential of the electrospinning hybrid membrane for MD. Su et al. [167] also prepared the PVDF-co-HFP membrane by electrospinning method and employed it in DCMD. The membrane showed a contact angle of 128° and a flux of 4.28 kg/m^2^. A total of 99.99% salt rejection was obtained when operating in MD for 12 h.

### 4.3. Preparation of PVDF-co-CTFE Membrane

#### 4.3.1. NIPS

The NIPS method is known as the dry–wet phase inversion method. Wang et al. [107] prepared a PVDF-co-CTFE flat sheet and hollow fiber membranes by the NIPS method. LiCl and PEG were utilized as pore formers and their effect on the membrane properties were also investigated. DMAc was used as a solvent and the membranes were characterized by SEM, XRD and FTIR, etc. The optimization of the membrane preparation conditions was investigated and applied in MD experiments. The final membranes showed the flux of 62.09 kg/m^2^h and the conductivity of the permeate side remained below 5 μS/cm.

#### 4.3.2. EIPS

Zheng et al. [58] prepared PVDF-co-CTFE membranes by EIPS and NIPS method. The polymer was mixed with additive and non-solvent to prepare the casting solution at the temperature of 30 °C. The solution was then stirred for 24 h to achieve the homogeneous transparent solution. After degassing under a vacuum oven for 24 h, the casting solution was spread uniformly on the support at the temperature of 50 °C in the oven. Then the solution was maintained in the oven for 15 s for evaporation. The membrane was then immersed in the non-solvent coagulation bath to form the membrane. Then the membrane was taken out from the coagulation bath for washing solvent and additives. The membrane was finally dried in the air. The effect of the LiCl additive was investigated on the membrane morphology and pore structure. The membrane preparation during phase inversion was also investigated for the membrane structure. The additive with 5% LiCl was tested in DCMD with a flux of 21.85 kg/m^2^ and salt rejection of more than 99.99%, and the conductivity was lower than 15 μS/cm. The membrane showed no wetting for about 6 h in MD.

### 4.4. Preparation of PVDF-co-TFE Membrane

#### NIPS + EIPS

Feng et al. [57] prepared PVDF-co-TFE membranes by the phase inversion method. The method used for phase inversion can be seen as a combination of the NIPS and EIPS methods. The polymer and additives were dissolved in the DMAc at the temperature of 50 °C. After the homogenous dope solution was prepared, the solution was degassed for 24 h at the temperature of 30 °C. The solution was poured on a glass and remained at this state for 30 s and then put into the distillated water for 10 min for forming of the membrane. Then the membrane was immersed into ethanol for 2 days to remove the remaining solvent and additives. Finally, the asymmetric porous hydrophobic membranes were obtained. The membranes were tested in MD and compared with the performance of a PVDF membrane. The mechanical strength and contact angle of the PVDF-co-TFE membranes were better than that of the PVDF membranes, indicating the potential as membrane materials for the MD process.

### 4.5. Preparation of PTFE Membrane

#### 4.5.1. Stretching and Sintering

Stretching and sintering is normally suitable for those polymer materials that cannot be fabricated by phase separation methods. Particularly PTFE, which is a typical crystalline polymer that has negligible solubility in all common solvents, and it has extremely high melting viscosity. A method involving mixing and aging, billet preforming, paste extrusion, stretching and sintering has been developed to fabricate a porous PTFE membrane. The preparation process is shown in the following Figure 21. Membrane porous structure is formed during the stretching process; therefore, the stretching step is critical for formation of pores. It influences the membrane’s mechanical strength and pore structures.

Li et al. [122] investigated and optimized the stretching conditions of the PTFE membrane preparation and its performance in MD. The result shows the highest peak in pore size distribution, when the stretching rate is 30%/s. At low stretching temperature and high stretching ratio, salt rejections were excellent up to 99.99%. Xiong et al. [168] also investigated PTFE ultrafine fibrous membranes by sintering before being electrospun with different PTFE/PVA mass ratios.

#### 4.5.2. ElPS

Khumalo et al. [169] prepared PVDF/PTFE flat sheet membranes by the EIPS method; the polymer was dissolved in NMP solvent and stirred to obtain the casting solution. After degassing, the membranes were casted on a glass with the thickness of 0.2 mm. Then the solution containing nanoparticles were casted on the membrane. The membranes were exposed to the air for evaporating the solvent by EIPS method. The membranes were formed at 40 °C and dried at 60 °C to remove the solvent completely. The membranes were tested in DCMD for removing urine components such as ammonia nitrogen, K^+^, Na^+^, TOC, etc. This shows that the MD technology could be used for water regeneration.

#### 4.5.3. Electrospinning

Xu et al. [170] prepared nanofiber PTFE membranes by the electrospinning method. The PTFE membrane was co-electrospun by polyacrylonitrile (PAN) on the PTFE precursor membrane. The Stőber method was chosen to grow SiO_2_ nanoparticles on the membrane surface and fix them on the PAN surface through chemical bonds by sintering. The SiO_2_ nanoparticles were then fluorinated by trimethoxy (1H, 1H, 2H, 2H-heptadecafluorodecyl) silane (17-FAS). The results show that the membrane had a water contact angle of 166.9° and oil contact angle of 134.5°. This amphiphobic membrane showed a stable flux of 17.09 L/m^2^h and high salt rejection of 99.96% in MD for a hypersaline solution containing oily saline and surfactant. It was endowed with high anti-scaling and anti-fouling properties and showed great potential in MD for wastewater treatment.

### 4.6. Preparation of FEP Membrane

#### Melt Spinning

Chen et al. [126] prepared FEP membranes by using melt spinning method. A mixture of FEP, DOP, KCl and SiO_2_ were spun into the hollow fibers in a certain weight ratio by a twin-screw spinning machine. Under certain conditions, the FEP hollow fiber membranes were prepared. The membranes were then applied in VMD and obtained the rejections of 99.9%. Huang et al. [171] also prepared FEP hollow fiber membrane by melt-spinning method and characterized them by porosity, LEP, hydrophobicity, pore size for MD.

### 4.7. Preparation of ECTFE Membrane

#### 4.7.1. TIPS Method

As indicated above, our group [128] prepared ECTFE flat sheet membranes by TIPS method by using a green solvent ATBC. The ECTFE particles was dissolved in ATBC solvent at 250 °C for 5 h to form a homogeneous solution and degassed for 4 h. The dope solution was poured into a preheated mould with a thickness of 250 μm for 20 min. Then the mould was put into a water bath for quenching at a certain temperature. Then the membrane was immersed into ethanol for 24 h to wash the remaining ATBC. The membrane was finally dried at 25 °C for 2 days. The effects of polymer concentration and quenching temperature on the membrane structure and performance were investigated. The membrane was finally applied in MD and achieved a flux of 22.3 L/m^2^h and a salt rejection rate of 99.9%.

#### 4.7.2. Dip-Coating Method

Our group prepared flat sheet ECTFE membranes on a non-woven support by dip-coating method [97]. Low point ECTFE particles were immersed in the first diluent DEA. DEG was chosen as the second diluent. The transparent homogeneous dope solution was obtained after stirring at 160 °C for around 1 h. Then after about 3 h of degassing, the non-woven support was immersed in this dope solution at the same temperature. The immersion time was investigated. After the immersion, the membrane was taken out and cooled in a DEG coagulation bath at 5 °C. The residue solvent was extracted by washing the membrane in 2-propanol and finally drying in the air. The membrane was applied in DCMD at 40–60 °C feed temperature, and a highest flux of 22 L/m^2^h was obtained and a highest 99.8% salt rejection was achieved, which shows the good performance of this type of non-woven supported membrane in MD.

#### 4.7.3. Low Temperature TIPS Method

The low temperature TIPS method is to prepare the membrane by TIPS method but at a lower temperature than its melting point. Polymer used in this method is a low melting point ECTFE polymer, which is also called Halar^®^ 901, supplied from Solvay Co., Ltd. Pan et al. [172] prepared ECTFE hollow fiber membranes using the L-TIPS method. The ECTFE particles were dried in a vacuum oven to remove moisture before usage. The particles and solvent were mixed at a certain proportion and then spun in the twin-screw machine to get the pristine hollow fiber membrane. N_2_ was used to form the lumen of the hollow fiber, and the membrane was entering into a quenching bath to form the membrane. The residue solvent was extracted by immersing the membrane into the ethanol and the ethanol was washed with pure water. The membrane was finally dried in the air. Effects of the additive on the membrane structure and morphology were investigated. For this type of preparation method, NIPS and TIPS methods are simultaneously used. The membrane exterior surface becomes thinner with the concentration of composite powder and shows a bicontinuous structure. The membrane was applied in a submerged vacuum membrane distillation (SVMD), and a flux of 10.8 L/m^2^h and a salt rejection of 98.9% was achieved.

Since most of the preparation methods of ECTFE membrane lie in using the TIPS method, the selection of solvent is an important issue for the TIPS method. The research direction of ECTFE is in developing new green solvents and modifying membranes with high characteristics including hydrophobicity, mechanical strength and fouling resistance [173,174]. Green solvents are not only suitable for preparing ECTFE membranes but can also be used for replacing toxic solvents of other membranes, which will be discussed further in the next Section 4.8.

### 4.8. Green Production of Fluoropolymer Membranes for MD/MCr

We listed some common preparation methods of fluoropolymeric membranes for MD/MCr. The polymers used for fabricating membranes have to be dissolved in solvents to form the liquid dope solution. Because of the toxic feature of the solvents used for membrane fabrication, such as N-Methyl-2-pyrrolidone (NMP), dimethylacetamide (DMAc), dimethylformamide (DMF), these commonly used solvents are harmful to humans if they are inhaled by breathing or touched by skin. These solvents are harmful to unborn babies and may cause serious eye irritations. They can also cause many serious diseases such as acute hepatocellular injury, steatosis and even testicular cancer, reported by the International Agency for Research on Cancer (IARC) [27,175]. Therefore, for reasons of worker safety and due to environmental problems, many researchers are now changing their research directions into more environmentally friendly solvents/diluents.

#### 4.8.1. Triethyl Phosphate (TEP)

TEP is a much safer solvent, and it is only harmful when swallowed. Researchers using TEP to fabricate PVDF flat sheet and hollow fiber membranes for MD. Nejati et al. [176] produced PVDF flat sheet membranes with porosity below 75% and has a flux of 40 L/m^2^h (LMH). Chang et al. [27] also produced PVDF flat sheet membranes using TEP as the green solvent with the highest flux of 13 LMH in DCMD. Chang et al. compared TEP/PVDF dope solutions with NMP/PVDF system and proved that when both the coagulant solution and the bore fluid contained 30 wt.% TEP, the prepared hollow fiber membranes presented a promising flux of 20 kg/m^2^h and LEP of 1.8 bar. Fadhil et al. [165] prepared novel PVDF–HFP flat sheet membranes with TEP as the green solvent and LiCl as the additive. By adding IPA in the coagulation bath, the prepared membranes have higher membrane porosity, larger pore size, higher contact angle and higher DCMD permeation at the temperature of 60 °C (16 kg/m^2^h) and the salt rejection at 99.3%.

#### 4.8.2. Ionic Liquids (ILs)

Ionic liquids (ILs) are a group of organic salts (containing anions and cations) that exist as liquids at a low temperature (below 100 °C). Because of its immeasurably low vapor pressure, it is called a green solvent. ILs has many superior characteristics, such as chemical and thermal stability, recyclability, non-volatility and high ionic conductivity [177,178]. It is attractive to researchers because it can dissolve some materials that are hardly soluble in other solvents and show different performance [179]. Typical ILs including [BMIM]Tf_2_N [180], [EMIM]Tf_2_N [181], [C_2_MIM]OAc [182], [EMIM]OAc [183], [MMIM]DMP, [EMIM]DEP, [EMIM]SCN [184], [BMIM]SCN [185], etc., were used as green solvents to prepare membranes, but most of them are used for gas separation and blended with CA membranes. For more applications in MD and the preparation of hydrophobic MF and UF membranes need to be further explored in the future. Jun et al. [186] prepared PVDF–PTFE membranes with the incorporation of ionic liquid [BMIM]PF6 as solvent. The incorporation of the ionic liquid decreased the pore size to about 0.58 μm and increased the LEP to 83 kPa. Additionally, the operation of the composite membrane in DCMD showed a longer lifespan, of about 26 h.

#### 4.8.3. PolarClean

PolarClean is an environmentally friendly solvent due to biodegradable properties [187]. It derives from 2-methylglutaronitrile (MGN), and it does not pose risks for human health and the environment. It has a high boiling point and is miscible with water. Hassankiadeh et al. [188] used PolarClean as a solvent to prepare PVDF hollow fiber membranes for the first time. The effects of PVP concentration, poly(methylmethacrylate)(PMMA) concentration, glycerol concentration, coagulation bath temperature on the membrane structures were investigated. The results show that PVDF/PolarClean membranes led to α- and β-phase polymorphisms and the PVP additives formed the β-phase on the membrane surface. Jung et al. [189] fabricated a PVDF hollow fiber membrane with PolarClean as solvent by NIPS–TIPS methods. The effects of polymer concentration, coagulation bath temperature and composition, and additives on the membrane morphologies were investigated. The results show that the membrane prepared with Pluronic as an additive achieved narrow pore size distribution and water permeability to 2800 L/m^2^hbar. Zou et al. [190] prepared a PVDF membrane with Polar Clean as a green solvent via the co-casting method; the prepared membrane had a salt rejection of 99.9% and a contact angle over 120°. The highest flux reached 37 kgm^−2^h^−1^. Saidi et al. [101] also prepared a PVDF–HFP flat sheet membrane by the NIPS method with Tamisolve^®^ NxG as the green solvent. The prepared membranes were utilized in DCMD and MCr tests. The membrane showed good performance compared to the commercial PVDF membranes. The rejection rates were higher than 99.9% with good trans-membrane flux.

#### 4.8.4. Organic Carbonates

Organic carbonates are also the sustainable solvent alternatives in membrane preparation. As an ester of carbonic acid, it consists of a carbonyl functional group attached to two alkoxy groups. General structures are R_1_-O(C=O)O-R_2_. It is biodegradable and has low eco-toxicity. It includes propylene carbonates (PC), ethylene carbonates (EC), glycerol 1,2-carbonate, dimethyl carbonates (DMC), diethyl carbonate (DEC), 1,2-Hexylene carbonate, and butylene carbonates (BC), etc. [191]. Ismail et al. [192] utilized EC, PC and BC as green solvents to fabricate PVDF membranes. Experimental and simulation tests were both investigated towards the phase inversion mechanism, crystalline structure, polymorphisms and morphologies, etc. The prepared PVDF membrane with EC as a green solvent showed the highest porosity and pure water permeability and therefore was applied in DCMD (with water flux at 25.6 kg·m^−2^·h^−1^). The membranes prepared from EC or PC were β-phase and the membranes fabricated from BC were the mixture of α-phase and β-phase. The results indicate that the cyclic organic carbonates are attractive green solvents.

#### 4.8.5. Other Non-Toxic Solvents

Other less toxic solvents such as γ-Valerolactone [193], diethylene glycol monoethyl ether acetate (DCAC) [194], glycerol triacetate (TRIACETIN) [144], triethylene glycol diacetate (TEGDA) [195], N,N-dimethyl lactamide [196], acetyl tributyl citrate (ATBC) [197], acetyl triethyl citrate (ATEC), triethyl citrate (TEC), diethyl adipate (DEA) [198], methyl lactate [199], maleic acid dibutyl ester (DBM) [200], dibutyl sebacate, γ-butyrolactone (γBL) [201], and dihydrolevoglucosenone (Cyrene^TM^) [202] have potential to be used as green solvents for MD.

Solvents including acetone, ethyl acetate, 2-methyltetrahydrofunan (2-MeTHF) were selected as prospective greener alternatives for tetrahydrofunan (THF) and dichloromethane (DCM) to dissolve polymer inclusion membrane (PIM) components. THF is commonly used to dissolve the base polymers PVC and PVDF-HFP, while DCM is used to dissolve cellulose triacetate (CTA). Carner et al. applied these greener solvents to fabricate PVDF-HFP-, PVC- and CTA-based PIMs and the results demonstrate that these PIMs can be performed in bio-based solvents instead of the conventional toxic solvents [203]. Russo et al. [155] used dimethyl isosorbide (DMI) as a green solvent to dissolve PVDF and PES to prepare UF and MF membranes by NIPS–VIPS method. Water permeability indicates that this membrane is suitable for water treatment. All these greener solvents have potential applications in MD and MCr processes, which could be used for future new membrane preparation.

## 5. Modifications of Fluoropolymer Membrane for MD/MCr

For the membranes used in MD and MCr, the higher their hydrophobicity, the stronger their anti-wetting abilities. The hydrophobicity can improve the stability of its long-term operation during the process and prolong the lifespan of the membrane in the device. The hydrophobicity of the membrane is determined by the surface energy and surface roughness of the membrane. Generally speaking, the lower the surface energy of the membrane, the weaker the adhesion of the liquid, therefore the higher the contact angle between the membrane surface and the liquid, and the higher the hydrophobicity [204]. According to the Cassie–Baxter equation, the hydrophobicity (contact angle) of the membrane could be related to the membrane surface roughness [205,206]. The increase of the surface roughness of the material will lead to the increase of the gas–liquid contact area and the enhancement of the interface effect. Therefore, introducing low surface energy substances is an effective way to improve the hydrophobicity of the membrane [207].

Common hydrophobic membrane modification methods include surface coating, surface grafting and blending, shown in Table 5.

Modification methods of the fluoropolymer membrane for MD/MCr are summarized in the following section.

### 5.1. Modification of PVDF Membrane

#### 5.1.1. Surface Coating

Coating is a very necessary technology to improve surface properties of the membrane. Especially for MD/MCr, improving the membrane hydrophobicity and surface roughness is one of the most common ways to enhance the membrane application for MD/MCr. It can be classified into dip coating and filtration coating. It can be described as a process whereby the membrane is immersed in the coating solution or liquid under a certain temperature and for a certain period of time. After immersion, the coated membrane is put into the oven and dried under a certain temperature for a certain period of time. The drawback of this method is that the coating layer is not quite stable compared with the grafting method, since the relatively weaker interactions of the coating layer can be washed by feed liquid during the application [210].

Our group coated Hyflon AD on the original PVDF membrane to increase its hydrophobicity and tested the composite membrane in VMD. The coating concentration, coating time, coating temperature and heat treatment time were tested. The results show that the performance of composite membrane is better than that of the original PVDF in VMD. Hyflon AD is a novel candidate for coating materials on PVDF membrane in VMD. The composite membranes were also tested in different feeds for a fouling experiment, which showed the promising stability of flux and salt rejection [130,211]. For MD, materials include SiNPs/polystyrene [212], TiO_2_ [213], SiO_2_ [214], fluorododecyltrichlorosilane (PFTCS) [215], ZnO [216], PFPE [94], semi-IPN hydrogel [217], polydopamine (PDA) [218], PDMS/CNT [219], 1-Butylpyridinium hexafluorophosphate (BPPF6) [220], β-FeOOH nanorods [221], zeolitic imidazolate frameworks (ZIFs) [222], aerogel [223], etc. were coated on PVDF membrane to optimize their performance.

#### 5.1.2. Surface Grafting

Surface grafting is a modification method on the membrane surface based on the connection of chemical bonds between membrane surface and the grafted materials, which leads to a more decent layer on the membrane surface. Grafting methods include chemical grafting, enzymatic grafting, gamma rays grafting, ultraviolet grafting, plasma grafting, etc. [224]. Most grafted methods were performed on the hydrophilic inorganic membrane with a hydrophobic layer [225,226]. Typical grafted materials include perfluoroalkylsilane (PFAS) [227], fluoroalkylsilane (FAS) [91], PDMS [228], poly(ethylene glycol) (PEG) [210], ethyl acrylate (EA) [229], etc. Han et al. [230] grafted sulfobetaine methacrylate (SBMA) on a hydrophobic PVDF membrane and compared the performance of the grafted membrane with the original membrane. The grafted membrane was conferred with anti-fouling and anti-wetting properties and had a lower contact angle than the pristine membrane. The grafted membrane had slightly higher flux in MD than the pristine membrane. The MD tests in oil emulsions for the grafted membrane had enhanced performance with respect to the pristine membrane. We grafted graphene on PVDF membrane to improve its properties. The preparation procedure is shown in Figure 22. The modified membrane showed an asymmetric structure and with finger-like and sponge pores. The membrane turned into a hydrophilic membrane after it was grafted with graphene. The membrane was applied in DCMD and showed high rejection and longer stability [93]. CF_4_ [231], TiO_2_ and PEG [210], multi-walled carbon nanotubes (MWCNT) [232], 1H, 1H, 2H, 2H-perfluorooctyltrichlorosilane [CF_3_(CF_2_)_5_(CH_2_)_2_SiCl_3_, PFOTS] [233], 1H, 1H, 2H, 2H-perfluorodecyl acrylate [234], SiNPs/APTES [235], ZnO [236], etc., are also grafted materials on the PVDF membrane surface for improving their properties in MD.

#### 5.1.3. Blending

Blending is one of the widely used methods to improve the membrane hydrophobicity during the preparation process of the membrane. It is normally by blending the membrane materials with the modified materials when preparing the dope solution. This method can not only improve the hydrophobicity property of the membrane, but also other parameters such as anti-fouling resistance, mechanical strength and membrane structures of the membrane. Common modification materials include nanoparticles, membrane materials, fluoropolymers, copolymers, etc., which are summarized in Table 6. Among all the blended materials, nanoparticles are attracting great interest in blending membrane modifications. They are commonly used in hydrophobic modification, and the hydrophobicity of the membrane is generally improved by increasing the surface roughness of the membrane. Inorganic nanoparticles such as TiO_2_, SiO_2_, CaCO_3_, etc., were used as blending materials in the fabrication of hydrophobic membranes [237]. Different to nanoparticles, fluoropolymer can increase the F/C ratio of the membrane surface, thus reducing the surface energy and improving the hydrophobicity of the membrane. The properties of the prepared membranes with blend polymers are often better than those of single polymer. As for the common copolymerization modification materials, PVDF copolymer is mainly represented, including P(VDF-co-HFP), P(VDF-co-CTFE), P(VDF-co-TrFE), etc. [238]. Through copolymerization, the crystallinity, melting point and glass transition temperature of PVDF can be changed, and finally PVDF copolymer is obtained, so it can meet the requirements of different membrane processes. Essalhi et al. [239] also blended fluorinated surface modifying macromolecule (SMM) into the casting solution by phase separation method. The SMM migrated to the membrane top surface during membrane formation process. The membrane was applied in DCMD and AGMD; the flux of the prepared hydrophobic/hydrophilic membrane reached 14.9 kg/m^2^h in AGMD and 2.7–3.3 times higher than AGMD in DCMD. Until now, PDMS [240], Cloisite15A^®^ clay (C15A) [241], GO [242], PVP and surface modifying macromolecules [243], LiCl/PEG/H_3_PO_4_/acetone [244], antimony-doped tin oxide (ATO) [245], ZnO [246], Fe_3_O_4_ [247], etc., have been blended with PVDF polymer for fabricating composite membranes for MD.

### 5.2. Modification of PVDF-co-HFP Membrane

#### 5.2.1. Surface Coating

Li et al. [257] prepared PVDF-co-HFP composite membranes by an in-situ heating method. The original PVDF-co-HFP membranes were prepared by electrospinning method, and then a series of Fe_3_O_4_ nanoparticle solutions were prepared. The composite membranes were prepared by vacuum filtration, which is one type of coating method. The membrane was dried at 70 °C for 6 h under 0.5 bar pressure. The composite membrane showed a flux of 0.97 kg/m^2^h and a rejection of 99.99% under solar irradiation in MD, which showing the light-to-heat conversion capacity of the Fe_3_O_4_ coating.

Up to now, TiO_2_ [258], Linde type L (LTL) zeolite NPs [259], 1H, 1H, 2H, 2H-perfluorodecyltriethoxysilane (FAS) [260], SiO_2_^@^PAN (HPi-SP) [261], PVDF [262], PDA-derived graphitic carbon spheres (GCSs) [263], carbon nanomaterials/powder-activated carbon [264], etc., have been coated on PVDF-co-HFP membrane surface to improve their performance in MD. Among all these materials, a TiO_2_ modified membrane [265] has been utilized in MCr for recovering NaCl and Na_2_SO_4_. Salt rejection obtained was 99.7% and flux reached 5.9 kg/m^2^h.

#### 5.2.2. Blending

Lu et al. [266] fabricated PVDF-co-HFP membrane blended with Fluorinated-decyl polyhedraloligomeric silsesquioxane (F-POSS) polymer by electrospinning method. The blending of F-POSS greatly increased the hydrophobicity of the membrane, lowering its surface energy. The membrane presented an omniphobic surface with the contact angle of 128.2° in ethanol. An MD test in a highly saline feed with low surface tension substances indicated that the flux and salt rejection maintained stability, which shows the great potential in desalination of the PVDF-co-HFP/F-POSS membrane in industrial wastewater.

At present, nanocrystalline cellulose (NCC) [267], silica nanoparticles (SiNPs) [98], LiCl [165], PVP, glycerol [268], DBP, PEG [269], MWCNTs [270], activated carbon [271], CaCO_3_, CaCl_2_ [272], ZnO [273], rGO [274], TiO_2_ [275], etc., have been utilized as additives during the preparation of PVDF-co-HFP membrane for MD.

#### 5.2.3. Surface Grafting

Ray et al. [104] prepared PVDF-co-HFP membrane and grafted cross-linked collagen on the surface to improve its mechanism and performance in DCMD. The membrane was prepared by phase inversion method by incorporating carbon nanofibers in the dope solution, and then started grafting. The composite membrane showed a contact angle of 113°, a salt rejection of 99.8% for 35 g/L NaCl solution as feed. It showed higher flux and a longer time in DCMD than the pristine membrane.

Other materials such as 3-Aminopropyl-triethoxysilane (APTES) [276] have also been used for grafting PVDF-co-HFP membrane surface for MD application.

#### 5.2.4. Blending + Surface Grafting

Chen et al. [277] fabricated superhydrophobic PVDF-co-HFP membrane with the mixture of reduced GO by electrospinning method and then grafted on the surface with fluoroalkylsilane POTS. The surface modification effects were investigated on the membrane morphology and physicochemical properties. The resulted membrane showed superhydrophobicity and tested in MD with NaCl as feed containing low surface tension surfactant SDS. The membrane exhibited high flux of 27.94 kg/m^2^h and salt rejection of 100%.

### 5.3. Modification of PVDF-co-CTFE Membrane

#### Blending

PVDF-co-CTFE membranes are easy to be grafted by atom transfer radical polymerization (ATRP) due to its C-Cl bond and without sacrificing its membrane stability, mechanical strength and thermal stability. It is normally grafted with some amphiphilic copolymer or zwitterionic groups. The resulting membranes normally present hydrophilic property and can be utilized for water treatment or gas separation, which is not in MD. Therefore, it is not easy to be surface-modified for MD.

Zheng et al. [58] prepared a PVDF-co-CTFE membrane by incorporating LiCl and PEG as additives to improve their performance in MD. Other additives were also blended with LiCl such as PVP, H_3_PO_4_, glycerol, and H_2_O. The results show that PVDF-co-CTFE composite membranes improved a lot when LiCl, PEG and H_3_PO_4_ were additives. The PVDF-co-CTFE membrane has great potential to be modified by different additives for MD. Mixed additives should be further explored in the future research for MD. Much research has also been conducted about LiCl additives in PVDF-co-CTFE membranes for MD application [278,279].

### 5.4. Modification of PVDF-co-TFE Membrane

At present, there is no reported modification of PVDF-co-TFE membrane for MD, which could be one of the future research directions.

### 5.5. Modification of PTFE Membrane

#### 5.5.1. Blending

Ju et al. [96] prepared a PTFE composite membrane by incorporating eight vinyl-grafted polyhedral oligosilicone (vinyl-POSS) nanoparticles into the nanofiber membrane, which was prepared by electrospinning method. The incorporation of the nanoparticles was utilized to increase the membrane porosity, surface roughness and mechanical strength. The concentration of the nanoparticles, preparation temperature and feed temperature in MD were investigated. The optimized composite membrane had a contact angle of 151 ± 4°. It presented a flux of 40 ± 2 L/m^2^h and a rejection above 99.99% in DCMD when feed and permeate temperatures were at 60/20 °C. It maintained 200 h stability in the long-term MD test.

Until now, GO [280], carbon nanotubes (CNTs) [281], PVA [282], polyethylene oxide (PEO) [283], have been incorporated into PTFE membranes for MD application.

#### 5.5.2. Surface Coating

Wang et al. [284] modified a PTFE membrane surface by electrospinning method with a hydrophilic coating of PVA, and then cross-linked with glutaraldehyde to improve its antifouling properties. The modified surface membrane presented an underwater oil contact angle of 148.7° and the substrate PTFE surface side with a contact angle in air of 134.5°. The composite membranes were tested in DCMD in saline water with 1000 mg/L crude oil emulsion and showed enhanced stable performance than the pristine membranes.

Coating material including dopamine (DA) [285], Fe_3_O_4_ and polyaniline (PANI) [286], GO [287], CNTs [288], P(PFDA-co-EGDA) [1H, 1H, 2H, 2H-perfluorodecyl acrylate (PFDA) monomer, ethylene glycol diacrylate (EGDA) cross-linker, and di-tert-butyl peroxide (TBPO) initiator] [289], polyimide (PI) [290], cellulose acetate (CA) and SiNPs [291] etc., have been coated on PTFE membrane surface for improving their properties in MD.

In fact, the electrospinning method is also a common way to carry out modification on PTFE membranes, which can be regarded as one of the coating methods, in principle.

#### 5.5.3. Plasma Treatment

Plasma treatment can be regarded as one of the grafting modification methods. Since there is only this type of method for grafting modification, it is specified here. Lai et al. [46] chose N_2_/H_2_ plasma treatment method to modify the surface of the PTFE membrane. The PTFE membrane consisted of a non-woven support and a dense PTFE layer. Different flow rates and power supply were investigated. After the treatment of plasma, the contact angles of the membranes were decreased. The fluxes in DCMD tests were increased but the salt rejections were expected to be sacrificed.

### 5.6. Modification of FEP Membrane

#### Blending

Chen et al. [127] prepared FEP membrane with SiO_2_ and DOP as additives and applied them in MD. The effects of these additives were investigated on membrane structure. The porosity and pure water flux were obviously enhanced. Rejection in VMD achieved 99.9%.

### 5.7. Modification of ECTFE Membrane

#### Surface Oxidation

The surface modification method can also be seen as one of the chemical modifications on a membrane surface. Anari et al. [292] utilized KMnO_4_ and HNO_3_ mixtures to modify a hydrophobic ECTFE membrane surface with hydrophilic appearance. The modification of the ECTFE membrane decreased the membrane thickness and therefore increased the trans-membrane flux. The hydrophilic surface of the modified membrane resulted in crystal deposition. To avoid the sacrifice of salt rejection of hydrophilic membrane in MD, they also performed a simple cleaning procedure for three cycles of MD.

### 5.8. Other Fluoro-Materials Modification Methods

In addition to the above-mentioned methods, sometimes different modification methods can be combined with each other and show better modification results. Other modification methods such as sol–gel methods, etc., are also promising hydrophobic modification methods for polymeric membranes. Some fluoro-materials can be coated or grafted on non-fluoro polymeric membranes (such as PP membranes), and the resulting composite membranes can be well-applied in the MD process.

Through the above modification methods and materials, many excellent hydrophobic membranes have been prepared and applied in MD and MCr. Shao et al. [211] was used to coat SiO_2_ particles on the surface of the previously hydroxylated PP membrane by sol–gel method, then fluorinated with fluorodecyl triethoxy silane (PFDTS) on SiO_2_ surface, so that fluorinated polymers were linked to SiO_2_ particles by hydroxyl groups in the form of chemical bonds. Because of the low surface energy of fluorinated polymers, the hydrophobicity of PP membrane was improved. The preparation process is shown in Figure 23. When the F/SiO_2_/PP-OH membrane was used to dispose high salinity NaCl (15 wt.%) and MgCl_2_ (3–9 wt.%) solutions in VMD, its permeability and separation performance remained stable for a long time (12 h).

## 6. Conclusions and Prospects

In this paper, fluoropolymer materials for MD and MCr processes are reviewed. The principles of MD and MCr were briefly introduced, and the preparation and modification methods of polymeric hydrophobic membrane were emphatically described. At present, researchers have prepared hydrophobic membranes with excellent performance by NIPS, TIPS, VIPS and other membrane preparation methods. In addition, by blending, coating, grafting or other modification methods, ideal membranes for MD and MCr application can be obtained. Research in this field is very active because the development of these processes requires further improvement of aspects such as the hydrophobic character of the membranes, improvement of their anti-fouling resistance and life span, and finally promotion of industrial scale application.

Among all the tested materials, P(VDF-co-TFE), P(VDF-co-TrFE), P(VDF-co-CTFE) and ECTFE are of particular interest. However, the available membranes still need further improvement, not only for the membrane preparation procedure, but also in their hydrophobic character. For P(VDF-co-CTFE) membranes, adding more additives in the membranes to improve their hydrophobicity could be one direction. For what concerns P(VDF-co-TFE) and P(VDF-co-TrFE), further investigation is necessary. Another aspect that requires in-depth research is the possibility of using green solvents instead of traditional, highly toxic solvents.

## Figures and Tables

**Figure 1 polymers-14-05439-f001:**
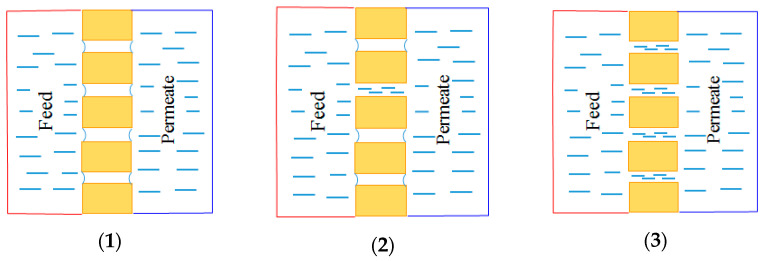
Mechanism of membrane wetting: (**1**) surface wetting, (**2**) partial wetting, (**3**) fully wetting.

**Figure 2 polymers-14-05439-f002:**
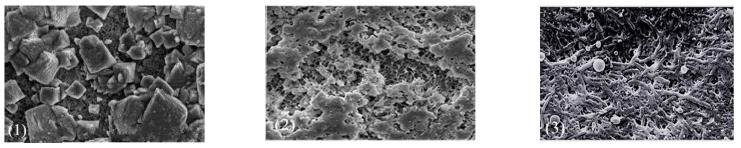
Fouling on membrane surface of MD: (**1**) inorganic fouling, (**2**) organic fouling, (**3**) biological fouling [17,18,19].

**Figure 3 polymers-14-05439-f003:**
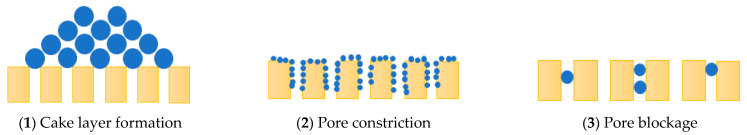
Mechanism of fouling. Reversible fouling: (**1**) Cake layer formation, (**2**) pore constriction. Irreversible fouling: (**3**) pore blockage.

**Figure 4 polymers-14-05439-f004:**
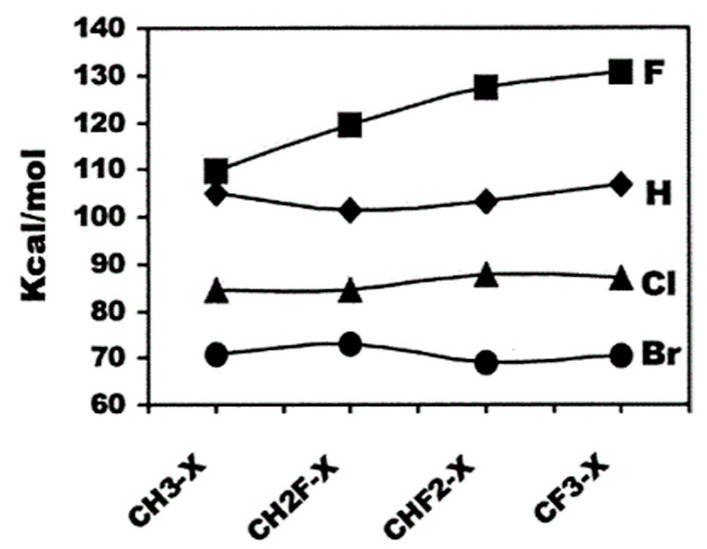
C-X bond energies in the model molecules CH_3_-X, CH_2_F-X, CHF_2_-X, and CF_3_-X, with X = H, F, Cl, and Br (reprinted from [25]).

**Figure 5 polymers-14-05439-f005:**
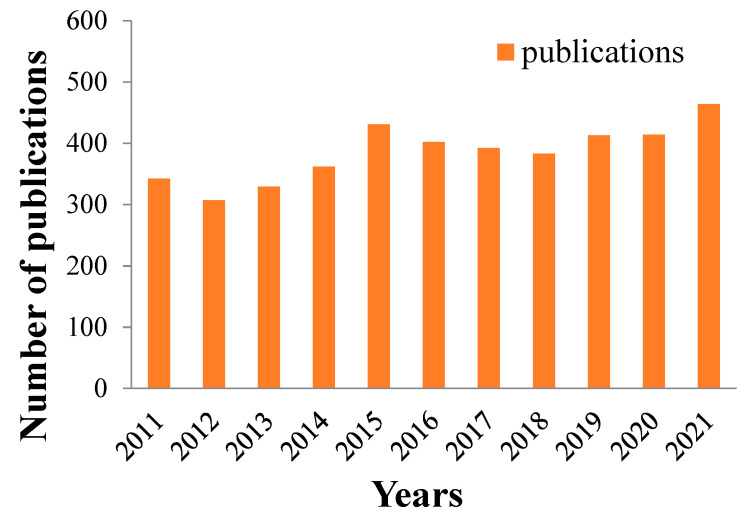
Number of publications of fluoropolymers applied in membrane distillation and membrane crystallization in the past 10 years in Web of Science (searched by using the key words “fluoropolymer membrane”, “membrane distillation” and “membrane crystallization”).

**Figure 6 polymers-14-05439-f006:**
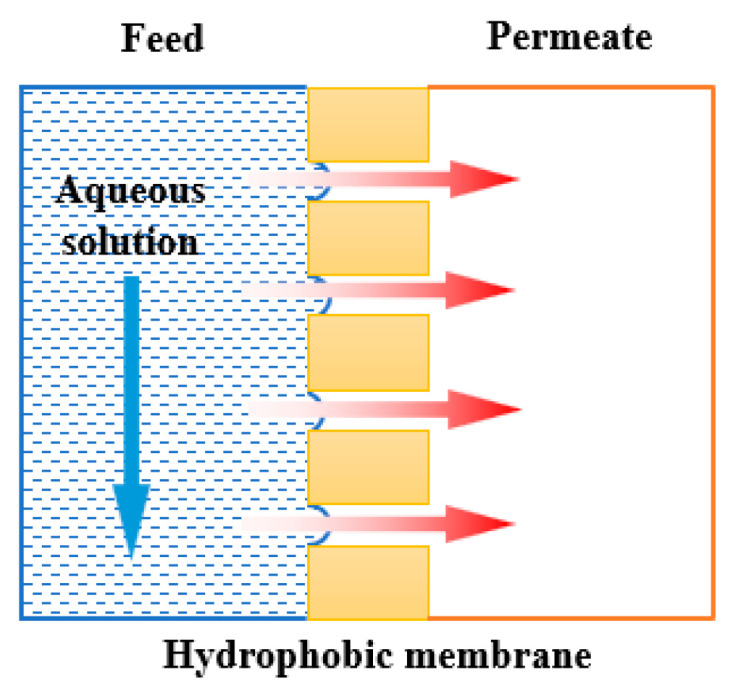
Principle of membrane distillation (MD) process.

**Figure 7 polymers-14-05439-f007:**
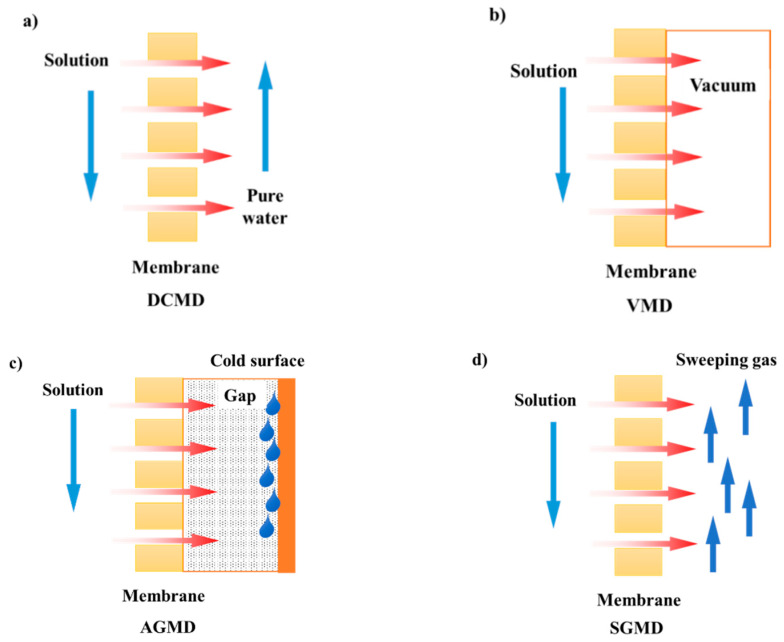
Main configurations of MD process: (**a**) direct contact membrane distillation (DCMD); (**b**) vacuum membrane distillation (VMD); (**c**) air gap membrane distillation (AGMD); (**d**) sweep gas membrane distillation (SGMD).

**Figure 8 polymers-14-05439-f008:**
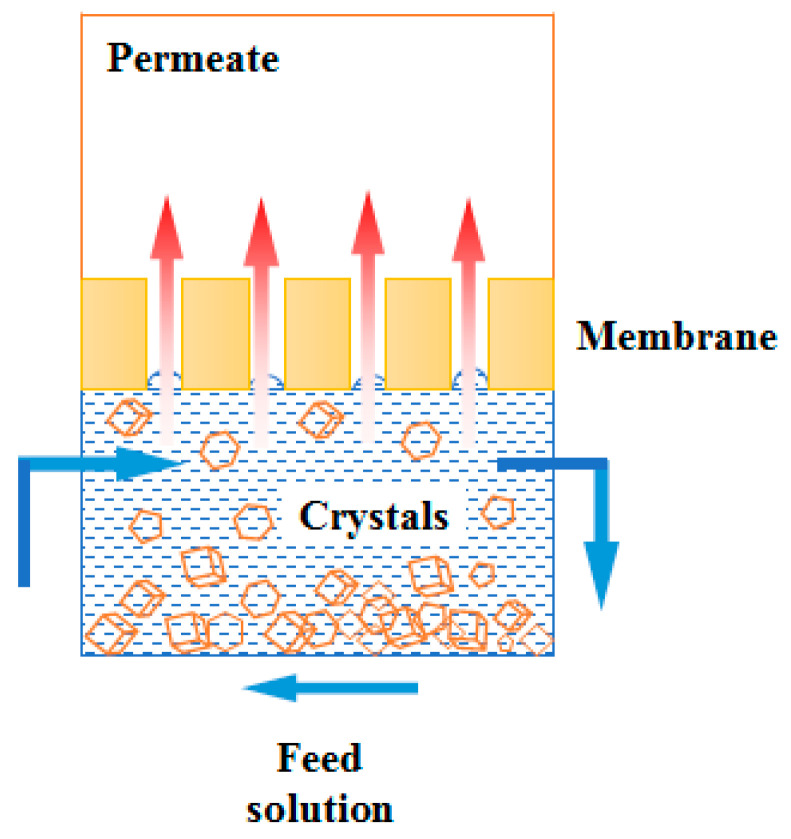
Principle of membrane crystallization process.

**Figure 9 polymers-14-05439-f009:**
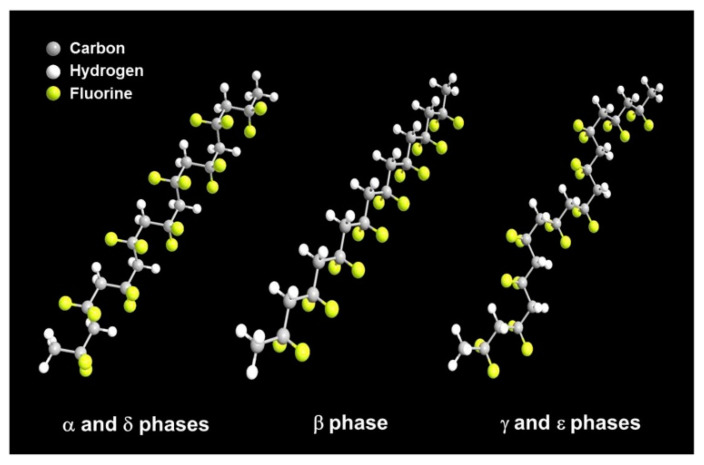
Conformations of five phases of PVDF molecular (reprinted from [51]).

**Figure 10 polymers-14-05439-f010:**
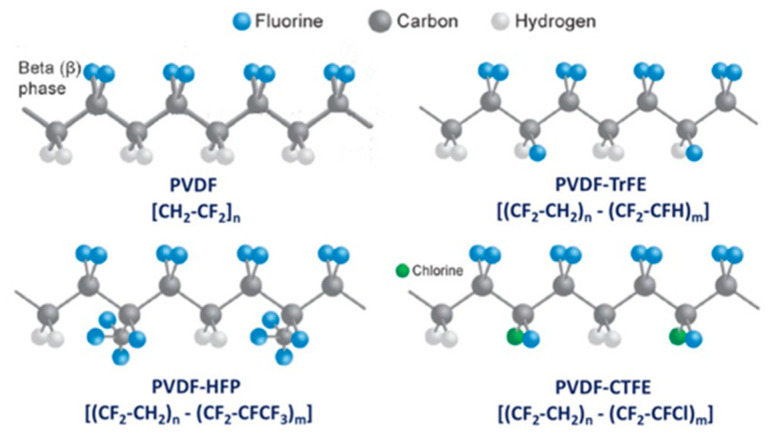
Structural configuration of PVDF and its copolymers: P(VDF–TrFE), P(VDF–HFP), and P(VDF–CTFE) (reprinted from [55]).

**Figure 11 polymers-14-05439-f011:**
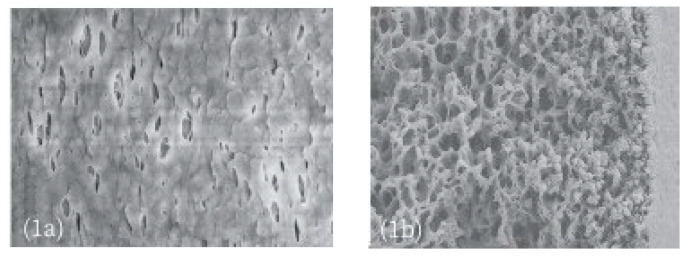
Surface (**a**) and cross-section (**b**) SEM images of (**1**) porous, (**2**) finger-like, (**3**) sphere-like pores and (**4**) ultrafine structure of PVDF membrane [45,103,104,105].

**Figure 12 polymers-14-05439-f012:**
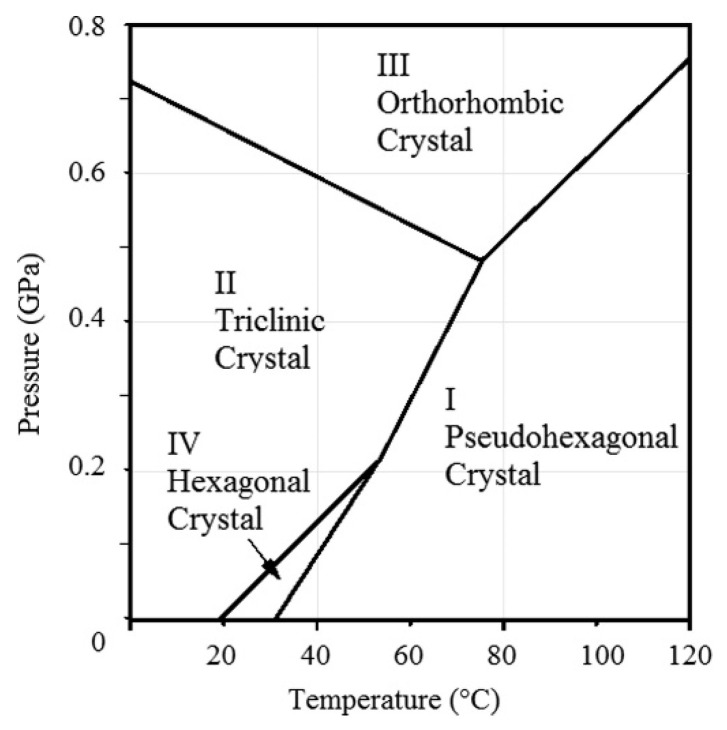
PTFE phase diagram at low pressure [117].

**Figure 13 polymers-14-05439-f013:**
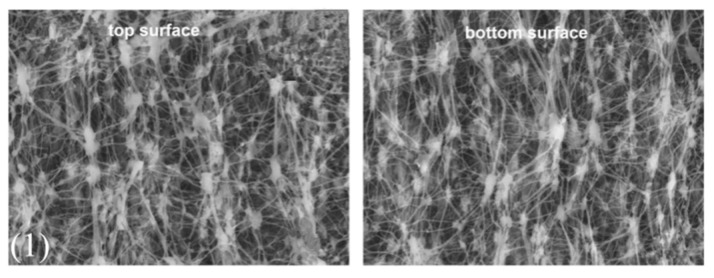
SEM images of different PTFE membrane with (**1**) flat sheet; (**2**) hollow fiber; and (**3**) ultrafine shape [119,121,122].

**Figure 14 polymers-14-05439-f014:**
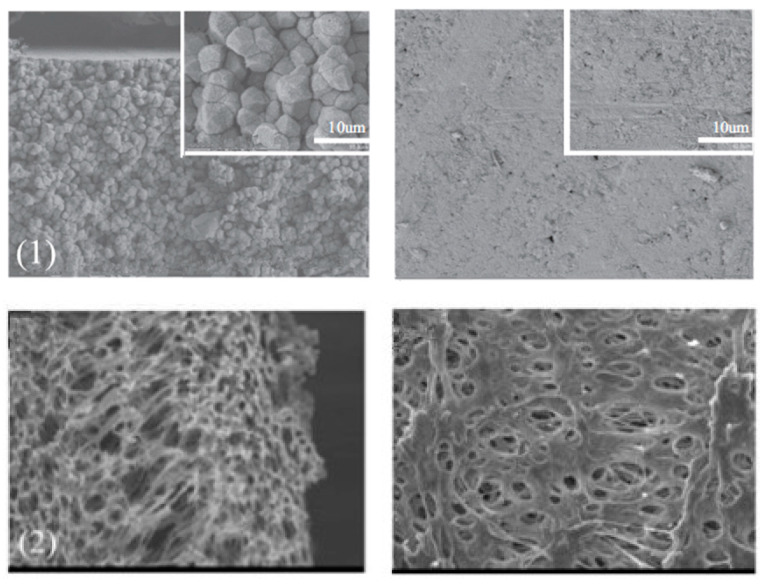
SEM images of cross-section and surface of (**1**) flat sheet; (**2**) hollow fiber ECTFE membrane [85,128].

**Figure 15 polymers-14-05439-f015:**
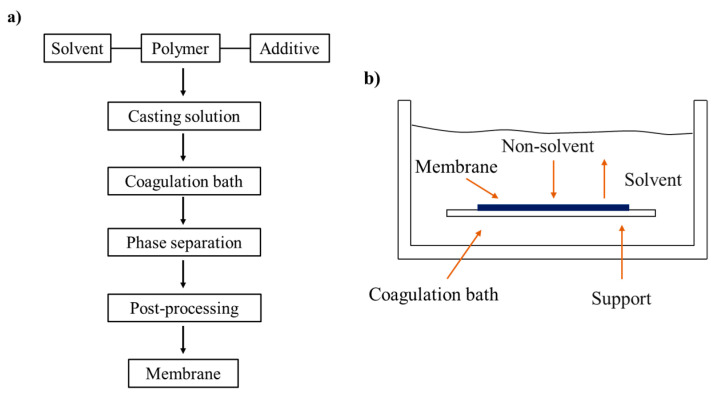
Process of NIPS method ((**a**): process of NIPS; (**b**): phase separation process).

**Figure 16 polymers-14-05439-f016:**
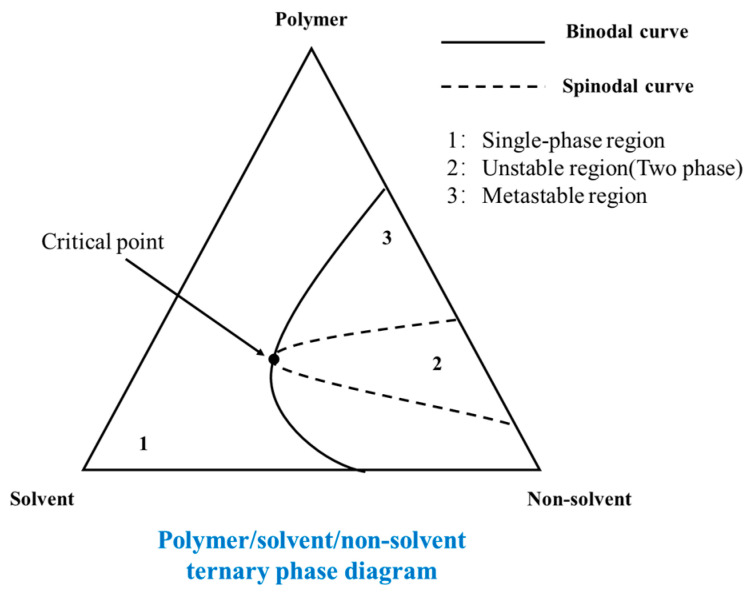
Phase diagram for NIPS method.

**Figure 17 polymers-14-05439-f017:**
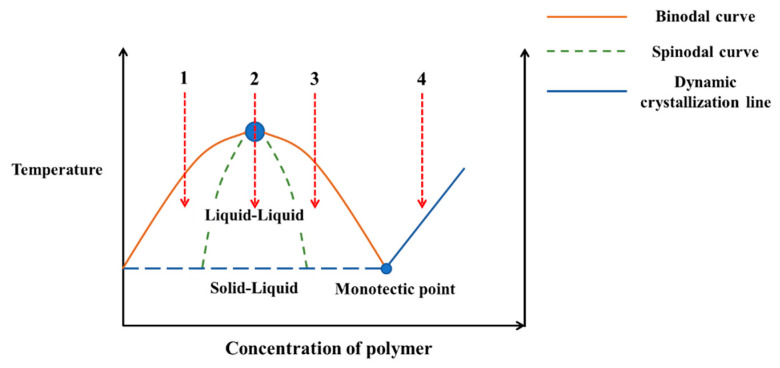
Phase diagram for TIPS method.

**Figure 18 polymers-14-05439-f018:**
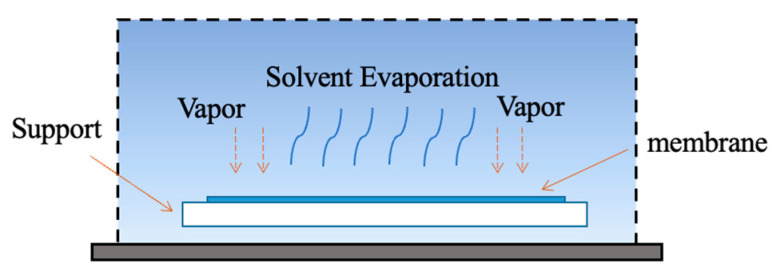
Process for VIPS method.

**Figure 19 polymers-14-05439-f019:**
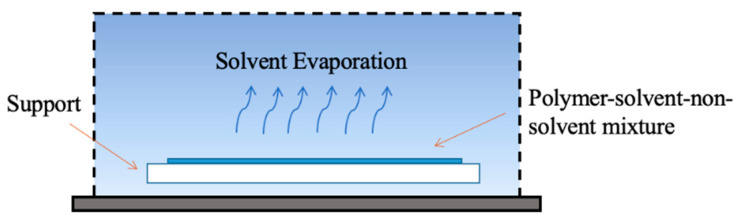
Process for EIPS method.

**Figure 20 polymers-14-05439-f020:**
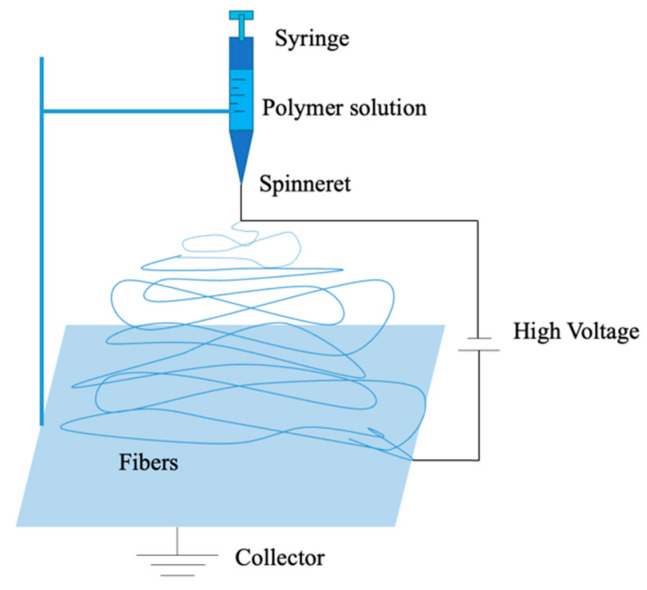
Set-up for electrospinning configuration.

**Figure 21 polymers-14-05439-f021:**
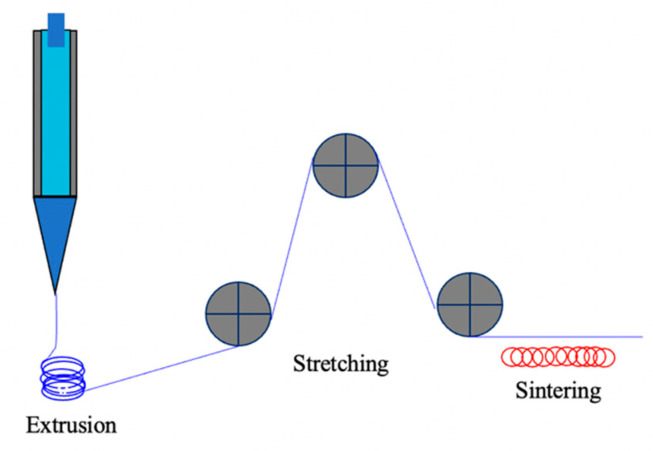
Process for stretching and sintering.

**Figure 22 polymers-14-05439-f022:**
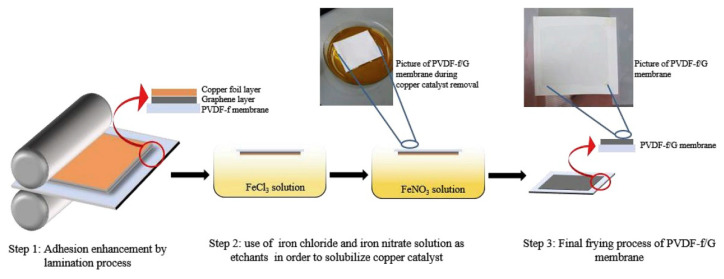
Procedures of grafting graphene on PVDF membrane (grafting) (reused from [93]).

**Figure 23 polymers-14-05439-f023:**
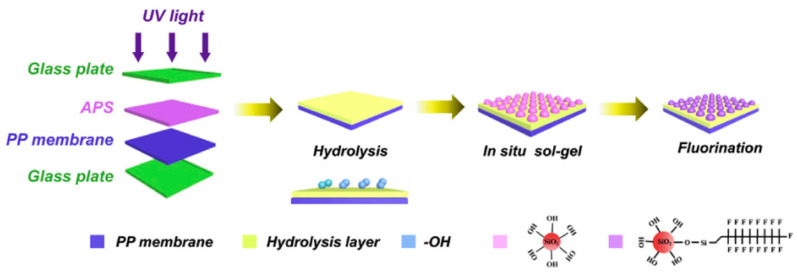
Schematic diagram for F/SiO_2_/PP-OH membrane preparation process (surface grafting) (reprinted from [211]).

**Table 1 polymers-14-05439-t001:** Main characteristics of MD and MCr.

Membrane Process	Configuration	Main Features	Application	References
MD	DCMD/OMD, VMD, AGMD and SGMD, etc.	Driving force: vapor pressure difference across membrane Objectives: Obtain high purity permeate, treatment of high concentration feed solutions. Advantages: be operated with low-grade waste heat sources	Seawater desalination, production of pure water, concentration of industrial wastewater such as biological solution, separation of azeotrope, etc.	[1,2,3]
MCr	DCMCr/OMCr, VMCr, AGMCr AGMD and SGMCr, etc.	Driving force: vapor pressure difference across membrane Producing macroscopic crystals with narrow size distribution and controlled morphology	Desalination and crystallization of salt solution, recovery of crystals from wastewater, preparation of biological macromolecules such as proteins and enzymes, etc.	[4,5,6,7,8,9]
	Solvent and anti-solvent crystallization	Concentration gradient/pressure gradient across the membrane	Pharmaceutical compounds, inorganic nanocomposites, organic salts	[10,11,12]

**Table 2 polymers-14-05439-t002:** Molecular formula of the most used hydrophobic fluoropolymer materials for MD and MCr.

Fluoropolymer Membrane Materials	Molecular Formula [48]
PVDF	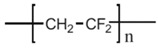
P(VDF-co-HFP)	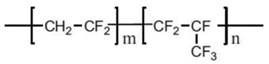
P(VDF-co-TFE)	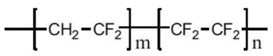
P(VDF-co-CTFE)	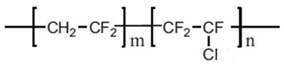
PTFE	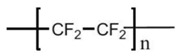
FEP	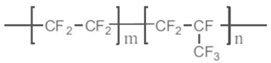
ECTFE	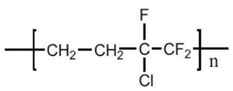

**Table 3 polymers-14-05439-t003:** Chemical properties of fluoropolymers for MD and MCr.

Polymers	Melting Temperature (T_m_) (°C)	Crystallization Temperature (T_c_) (°C)	Crystallinity Degree	Average Molecular Weight (Mw) (kg/mol)
PVDF [70,71]	110–171	50–140 (β phase lower temperature, α phase higher temperature)	0.09–0.6	600–700
P(VDF-co-TFE) [57]	<50	---	---	1570
P(VDF-co-CTFE) [72,73,74]	110–168	123–129	0.13–0.29	270–290
P(VDF-co-HFP) [75,76,77]	117–147	106–138	0.01–0.33	115–600
PTFE [78,79,80,81]	325–342	296–315	0.33–0.54	260–45,000
FEP [82,83,84]	147–280	189–234	0.37–0.4	250–600
ECTFE [85,86,87]	175–285	190–222	0.25–0.3	---

**Table 4 polymers-14-05439-t004:** Main applications of fluoropolymeric membranes in MD and MCr in recent years.

MD Type	Membrane Type	Preparation Methods	Contact Angle (°)	Feed Solution	Temperature (°C) T_f_/T_p_	Flux (LMH)	Rejections (%)	References
DCMD	Flat sheet PVDF	NIPS and VIPS	130.3	Sea water	60/20	23.5	>99.7	[91]
DCMD	Flat sheet PVDF	Commercial	124	Real wastewater with antibiotics	60/20	19.76	100	[92]
DCMD	Flat sheet PVDF	Commercial functionalized with graphene	73	0.5 M NaCl	70/20	19	99.9	[93]
VMD	Flat sheet PVDF	UV-curing	162.6	0.6 M NaCl	55/5	35	99.99	[94]
AGMD	Nanofiber PVDF	Electrospinning	130	0.6 M NaCl	(37–82)/22	11–12	98.7–99.9	[87]
SGMD	Flat sheet PVDF	VIPS and NIPS	138.2	0.5 M NaCl	25/20	8.6	---	[95]
DCMD	Nano fiber PTFE	Electrospinning	151	0.6 M NaCl	60/20	40	99.99	[96]
DCMD	Flat sheet ECTFE	TIPS	100–119	0.6 M NaCl	(40–60)/14	6.5–22	99.91–99.82	[97]
DCMD	P(VDF-co-HFP)	Electrospinning	150	0.6 M NaCl	(40–80)/20	48.6	99.99	[98]
DCMD	P(VDF-co-TFE)	NIPS	86.8	0.3 M NaCl	55/25	7.3	---	[57]
DCMD	Flat Sheet P(VDF-co-CTFE)	NIPS	97.28	0.6 M NaCl	55/25	20.65	99.95	[99]
DCMCr	Flat sheet PVDF	Commercial with coating	137	5.3 M NaCl	34/10.5	1.78–2.54	---	[6]
DCMCr	Hollow fiber PTFE	Commercial	108	Saturated thiourea solution	39/(24–39)	---	---	[100]
DCMD and MCr	Flat sheet P(VDF-co-HFP)	NIPS	90.18	0.05 and 5.3 M NaCl	36/15.5	0.62	>99	[101]
VMDCr	P(VDF-co-HFP)	TIPS	---	sub-soil brine with CaSO_4_ and NaCl	60	14	---	[102]

**Table 5 polymers-14-05439-t005:** Methods for hydrophobic modification on membranes for MD/MCr.

Hydrophobic Modification	Features
Surface coating	It mainly includes dip coating and filtration coating. The difference is that the dip coating is to immerse membrane into modified solution, so that the modified material forms a hydrophobic coating on membrane surface to improve the hydrophobicity; and the latter deposits the modified materials (generally nanoparticles) onto membrane surface by means of filtration property of membrane The properties of the modified membranes depend on the support layer, the composition of the coating solution, and the reaction conditions of the coating process
Surface grafting	Active site formed on membrane surface by oxidizing agent, irradiation, plasma, photoinitiation, etc., of which the modified polymer is a graft polymerized on the surface of membrane The grafted polymer is bonded to membrane surface in the form of a chemical bond, so that the modification effect is more remarkable, and the modification only occurs on membrane surface without affecting the properties of the membrane material [208,209]
Blending	Mixing membrane material with the modified material (or membrane materials) to prepare the casting solution The prepared blending membrane not only can improve the hydrophobicity, but also can improve the strength, anti-fouling, antibacterial property, etc.
Other methods	Copolymerization modification, membrane material is modified by introducing chemical group or molecular chain to the side or main chain of membrane material, and then prepare the corresponding casting solution Sol–gel method, substance containing high chemically active ingredient used as precursor, then uniformly mixed in the liquid state, and a sol would be produced by chemical reaction such as hydrolysis or condensation. Then, the sol is gradually aggregated between the aged rubber particles to form a three-dimensional network structure filled with non-flowing solvent. Finally, the gel forms a nanostructured material by drying or sintering

**Table 6 polymers-14-05439-t006:** Materials for modification of fluoropolymeric membrane for MD.

Materials		Features
Particle [248,249,250]	Nanoparticles such as SiO_2_, TiO_2_, ZnO_2_, CaCO_3_, etc. Membrane materials particles such as PVDF, PTFE, etc.	The presence of nanoparticles improves hydrophobicity of membrane by increasing the surface roughness of membrane The uneven distribution or agglomeration of particles may cause blockage of pores and have a certain influence on the permeability of the membrane
Fluoropolymers [132,251,252]	Fluoroalkylsiloxane (FAS), Fluorocarbon surfactant (FS), Hyflon, Teflon, etc.	The fluoropolymers form hydrophobic coating on the surface of membrane, reducing the surface energy of membrane to achieve the purpose of improving the hydrophobicity of the membrane The long-term stability of the modified hydrophobic coating plays a decisive role in the MD and MCr process
Copolymer [99,253]	P(VDF-co-HFP), P(VDF-co-CTFE), P(VDF-co-TrFE), PVDF-g-PSSA, etc.	Compared with the original membrane material, the copolymer has a certain degree of improvement in crystallinity, melting point and stability Significantly, the copolymer is more hydrophobic [238]
Other materials [254,255,256]	Polydimethylsiloxane (PDMS), Graphene, two-dimensional materials, additives, pore formers such as PVP, LiCl, PEG, etc.	Increase porosity, mechanical strength and good membrane structure, etc.

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
