# Peer review of "Fluoropolymer Membranes for Membrane Distillation and Membrane Crystallization"

_polymers, 2022, doi:10.3390/polym14245439_

Round 1

Reviewer 1 Report

I really liked this review. The value of review articles is that the authors provide data on the subject of the review from a large number of relevant works. In this review, about 300 articles were cited, that is, the authors did a great job. Various mechanisms are considered that reduce the hydrophobicity of fluoropolymer membranes, which affects the efficiency of mass transfer and separation. In fact, this review is devoted to various methods for increasing the hydrophobicity of fluoropolymer membranes, which will increase their service life. I suggest publishing this review as is. I did not find any inaccuracies or errors in this review.

Author Response

Thank you very much for the kind comments.

Reviewer 2 Report

Title: Fluoropolymer membranes for membrane distillation and membrane crystallization

Recommendation: Minor revisions needed as noted.

There are a few questionable points that the authors should try to more clearly address.

1. Is there any similar article published in the literature ?

1. Try to make more comprehensive abstract that should not be too long.

2. The abbreviation definition should be centered are justified.

3. Carefully check the superscript and subscript throughout the manuscript.

4. Need careful revision of the manuscript for grammatical structural and typo mistakes.

5. References should be in same style.

Author Response

Thank you very much for the comments. To respond your comments, please find the following response.

  1. Is there any similar article published in the literature?

-To my best knowledge, there is no similar review relating to my title.

  1. Try to make more comprehensive abstract that should not be too long.

-The abstract has been modified as follows:

Abstract: Fluoropolymer membranes are applied in the membrane operations such as membrane distillation and membrane crystallization. In these two membrane processes, the where hydrophobic porous membranes acts as a physical barrier separating two phases. Due to the hydrophobicity nature, only gaseous molecules are allowed to pass through membrane and are collected on the permeate side, while the aqueous solution cannot penetrate. Ultimately the purification and separation of some effective components in the feed side can be achieved. However, membrane distillation and membrane crystallization these two processes also suffer from problems as other membrane operations such as membrane wetting, membrane fouling or membrane scaling. Membrane wetting is a common and undesired unwanted phenomenon during membrane operations, which is caused by the loss of hydrophobicity of the porous membrane employed. It will Thisgreatly affects the mass transfer efficiency and separation efficiency. Simultaneously, membrane fouling occurs would happen along with membrane wetting and scaling, which will greatly reduces the lifespan of the membranes. Therefore, how strategies to improve the hydrophobicity of membranes has have been widely concerned and investigated by researchers. In this direction, hydrophobic fluoropolymer membrane materials are employed more and more for membrane distillation and membrane crystallization thanks to their high chemical and thermal resistance. This paper summarizes different preparation methods of these fluoropolymer membrane, such as non-solvent induced phase separation (NIPS), thermally induced phase separation (TIPS), vapor induced phase separation (VIPS), etc. Hydrophobic modification methods, including surface coating, surface grafting and blending, etc., are also introduced. Moreover Meanwhile, the research advances on the application of less toxic solvents for preparing these membranes are herein also reviewed.

This review aims to provide guidance to give an indication for researchers, for their future membrane development in membrane distillation and membrane crystallization, using fluoropolymer materials.

  1. The abbreviation definition should be centered are justified.

-The abbreviation has been changed in the manuscript and marked in red;

  1. Carefully check the superscript and subscript throughout the manuscript.

-Superscript and subscript have been carefully checked and marked in red;

  1. Need careful revision of the manuscript for grammatical structural and typo mistakes.

-Gramatical and typo mistakes have been revied in the text and marked in red;

  1. References should be in same style.

-Referene type has been changed in the reference and in the text.